# Seaweeds and Corals from the Brazilian Coast: Review on Biotechnological Potential and Environmental Aspects

**DOI:** 10.3390/molecules28114285

**Published:** 2023-05-23

**Authors:** Gustavo Souza dos Santos, Thais Luz de Souza, Thaiz Rodrigues Teixeira, João Pedro Cezário Brandão, Keila Almeida Santana, Luan Henrique Santos Barreto, Samantha de Souza Cunha, Daniele Cristina Muniz Batista dos Santos, Conor R. Caffrey, Natan Silva Pereira, Aníbal de Freitas Santos Júnior

**Affiliations:** 1Department of Life Sciences, State University of Bahia, Salvador 41150-000, BA, Brazil; keilaalmeidasantana@gmail.com (K.A.S.); luanbarreto8075@gmail.com (L.H.S.B.); 2Department of Analytical Chemistry, Chemistry Institute, Federal University of Bahia, Salvador 40170-115, BA, Brazil; tls2605@gmail.com (T.L.d.S.); dcmsantos@ufba.br (D.C.M.B.d.S.); 3Center for Discovery and Innovation in Parasitic Diseases, Skaggs School of Pharmacy and Pharmaceutical Sciences, University of California San Diego, La Jolla, CA 92093, USA; trteixeira@health.ucsd.edu (T.R.T.); ccaffrey@health.ucsd.edu (C.R.C.); 4Department of Exact and Earths Sciences, State University of Bahia, Salvador 41150-000, BA, Brazil; j.pedro.98@hotmail.com (J.P.C.B.); samsccunha.99@gmail.com (S.d.S.C.); nspereira@uneb.br (N.S.P.)

**Keywords:** Brazilian shoreline, seaweeds, corals, secondary metabolites, bioactivity, nutrients, potentially toxic elements, environmental

## Abstract

Brazil has a megadiversity that includes marine species that are distributed along 800 km of shoreline. This biodiversity status holds promising biotechnological potential. Marine organisms are important sources of novel chemical species, with applications in the pharmaceutical, cosmetic, chemical, and nutraceutical fields. However, ecological pressures derived from anthropogenic actions, including the bioaccumulation of potentially toxic elements and microplastics, impact promising species. This review describes the current status of the biotechnological and environmental aspects of seaweeds and corals from the Brazilian coast, including publications from the last 5 years (from January 2018 to December 2022). The search was conducted in the main public databases (PubChem, PubMed, Science Direct, and Google Scholar) and in the Espacenet database (European Patent Office—EPO) and the Brazilian National Property Institute (INPI). Bioprospecting studies were reported for seventy-one seaweed species and fifteen corals, but few targeted the isolation of compounds. The antioxidant potential was the most investigated biological activity. Despite being potential sources of macro- and microelements, there is a literature gap regarding the presence of potentially toxic elements and other emergent contaminants, such as microplastics, in seaweeds and corals from the Brazilian coast.

## 1. Introduction

Marine organisms are promising targets for prospecting chemical species with biotechnological applications. Among them, seaweeds and corals are recognized as prolific sources of novel molecules, with applications in the pharmaceutical, cosmetic, and chemical industries.

Previous reports have addressed the biological and chemical potential of seaweeds [1,2,3,4]. In recent decades, global seaweed aquaculture has produced more than 30 million tons of fresh biomass, with an estimated market value of more than $13.3 billion. *Laminaria*/*Saccharina* (35.4%), *Kappaphycus* and *Eucheuma* (33.5%), *Gracilaria* (10.5%), *Porphyra* (8.6%), and *Undaria* (7.4%) account for more than 95% of global seaweed farming production [5]. Therefore, seaweeds are part of a promising millennium market, providing several ecological, social, and economic benefits [5,6]. Coral reefs are one of the most endangered environments on the planet, having lost up to 67% of their historic global range. Furthermore, as documented in a recent report by the Global Coral Reef Monitoring Network, the remaining intact coral reef systems are under threat from anthropogenic actions and climate change effects [7,8]. Previous reports in the literature indicate the potentialities of the secondary metabolites of coral, including cytotoxic, anti-inflammatory, antifouling, and anti-microbial effects, in addition to bone repair and neurological benefits [9,10,11,12].

The UN Decade of Ocean Science for Sustainable Development 2021–2030 Implementation Plan (IOC-UNESCO 2020) discusses the structuring of ‘Blue Economy’ actions, which came to the fore during the preparations for the Rio + 20 or Earth Summit (UNCSD 2012). Since then, many coastal and island nations have expressed concern about the neglect of the ocean’s role in the economic and cultural lives of hundreds of millions of the world’s poorest and most vulnerable people of coastal and island nations [13,14].

Despite their biotechnological potential, marine organisms suffer the ecological pressures derived from a rapidly changing climate [15] and anthropogenic actions [16], such as overfishing and pollution, which might lead to the bioaccumulation of potentially toxic elements and microplastics [17], as well as changes in the trophic structure. In recent years, the new Coronavirus (SARS-CoV-2) pandemic has stimulated the development of studies to map and monitor human-ocean interactions in near real-time during Coronavirus Disease 2019 (COVID-19), but two main problems have been highlighted: (1) manual and isolated data collection and processing, and (2) reliance on marine professionals for observation and analysis [18]. Social isolation and confinement reduced the impact of human activities and contact with marine organisms and, therefore, reduced the damage to the environment. On the other hand, restrictions and confinement may have impacted marine research development.

Due to its extensive shoreline and the status of a megadiverse country, Brazil holds the potential to be a reference lead country in marine biotechnology. In this context, this review describes the current status of the biotechnological and environmental aspects of seaweeds and corals in a comprehensive way, including publications from the last 5 years, that were produced in Brazil.

For this investigation, we carried out an exploratory search using the PubChem (https://pubchem.ncbi.nlm.nih.gov, accessed on 28 April 2023) and PubMed (https://pubmed.ncbi.nlm.nih.gov, accessed on 28 April 2023), databases from the National Center for Biotechnology Information (NCBI) of the United States of America; ScienceDirect (https://www.sciencedirect.com, accessed on 28 April 2023), a database operated by the Anglo-Dutch publisher Elsevier, and Google Scholar (https://scholar.google.com, accessed on 28 April 2023).

The search was completed in April 2023 using the terms Brazilian shoreline; seaweeds; corals; secondary metabolites; bioactivity; nutrients; and potentially toxic elements. Publications dating from January 2018 to April 2023, which were produced in Brazil, were included. The Boolean operator “AND” between the terms provides the intercession, restricting the scope of the search in the title, abstract, and keywords in order to obtain a more comprehensive and effective outcome for the search results. Six researchers searched through the literature and collected the data. All researchers revised the included articles. No automatic tools were used in the screening. The following were excluded: articles not in the English language, letters to editors, and papers that did not include any biological, chemical, nutritional, or environmental aspects of seaweed and corals performed outside of Brazil. This data was collectively stored and classified (using Microsoft Excel 2016) according to the year of publication, the type of study, the country of the corresponding author, and biological, chemical, nutritional, and environmental aspects. After selecting the included papers, the full text of the articles was analyzed and the articles were grouped for the discussion below, and included biotechnological, nutritional, and environmental aspects.

A technological prospectus for patents in the area of biotechnology was carried out, focusing on the use of algae and corals from the Brazilian coast for biotechnological purposes such as the development of drugs and cosmetics, the investigation of nutritional potential, or other commercial applications. Also, a search took place in the Espacenet database (https://worldwide.espacenet.com, accessed on 15 March 2023), maintained by the European Patent Office (EPO), using the keywords seaweeds; corals; biotechnology; medicinal; pharmaceuticals; cosmetics; and nutritional in the titles or in the abstract of the patent document. A combination of Boolean operators (AND, OR, and NOT) was used as selection filters to meet the inclusion criteria, namely patents granted between January 2018 and December 2022. We excluded patents that do not use algae and corals from the Brazilian coast for biotechnological purposes, and those not deposited in Brazil. In addition, a search was also carried out at the National Property Institute (INPI) (https://busca.inpi.gov.br/pePI, accessed on 15 March 2023), which has patent documents posted in Brazil, in which the search took place using the same cited keywords, either in the titles or in the abstract of the patent document, as well as by applying a combination of Boolean operators (AND, OR, and NOT), and the following inclusion criteria were used: patents published between January 2018 and December 2022. All of the compound structures were manually drawn using the ChemDraw Professional 16.0 software. 

## 2. Results and Discussion

### 2.1. Seaweeds along the Brazilian Coast: Bioactive Compounds and Microelements

Marine organisms have significant biotechnological potential, such as the synthesis of bioactive molecules, the production of biofuels, cosmeceuticals, nutraceuticals, and biopolymers. They have been engineered for biomedical purposes, the degradation of pollutants, and are used as biosensors for environmental quality [19]. Among them, seaweeds have historically been used as sources of components with industrial applicability. Agar extracted from red algae, such as *Gelidium* sp., *Glacilaria* sp., and *Pterocladiella* sp. has been widely used in the food industry as a thickening agent and a substitute for gelatin. In the pharmaceutical industry, its more purified form is used for separation apparatuses in molecular biology, whether in electrophoresis, immunodiffusion, or gel chromatography [20,21]. Carrageenan from red algae has 20% of its production go towards the manufacturing of skin care products, due to its thickening, gelling, emulsifying, and stabilizing properties [22]. The species of brown algae, predominantly *Laminaria japonica*, also has industrial applicability as a stabilizer for emulsions and foams, an encapsulation agent, and an agent for forming films and synthetic fibers, among other possibilities [23]. In addition, alginate’s gel-forming capacity and biocompatibility make it suitable for biomedical applications, such as the preparation of bandages, tissue engineering, and drug administration [24]. Polysaccharides present in *Ulva* spp. species have significant potential in the food industry such as new functional foods or food additives [25].

Marine biotechnology is globally recognized as an engine of economic growth. However, the field is mainly concentrated in the European Union (EU), North America, and the Asia-Pacific. Renowned marine biotechnology centers are in China (e.g., the Institute of Oceanology), Japan (e.g., the Shimoda Marine Research Center), the United States (e.g., the Scripps Institute of Oceanography), and Australia (e.g., the Australian Institute of Marine Science). In South America, marine biotechnology is an emerging and unexplored field [26]. Brazil has a coastline of almost 8000 km, which makes it one of the world’s longest, with an equatorial, tropical, and subtropical climate. Brazil is one of the world’s most megadiverse countries due to its extensive coastline, which is home to a number of endemic and well-adapted marine species [27]. This megadiversity can be translated into aggregate biotechnological applications with the potential to drive the new bioeconomy, which should lead the country to sustainable development in the next few years. The collection spots of the seaweeds and corals collected along the Brazilian coast and that were investigated with regard to their biotechnological composition in the period ranging from 2018 to 2022 are shown in Figure 1. 

Herein, we summarize the occurrence of secondary metabolites and their bioactivities in seaweeds and corals from the Brazilian coast, as well as their nutritional potential regarding the presence of macroelements. Reports addressing the bioactive secondary metabolites from Rodophyta algae were the most abundant, accounting for 37 investigated species, followed by Ochrophyta with 27 and Chlorophyta with seven investigated species (Figure 2).

The Rhodophyta species were mainly evaluated regarding their antioxidant and antiparasitic potential towards *Schistosoma mansoni* and *Trypanosoma cruzi*, while in the Ochrophyta algae, the major investigated bioactivities were antioxidant and antiviral, including the Zika, Chikungunya, and HIV viruses. Reports addressing the bioactivity of the Chlorophyta species were mainly focused on the antioxidant potential, as shown in Figure 3. 

#### 2.1.1. Ochrophyta

Brown algae are the most abundant class of marine macroalgae, and despite their recognition as a kelp-formation species that occurs mainly in cold and temperate regions, there are approximately 165 species registered in the western South Atlantic [28]. 

In this study we report a series of terpenoids that were isolated and putatively annotated from the *Dictyota*, *Canistrocarpus*, and *Stypopodium* genera (Figure 4, Table 1). Additionally, monogalactosyldiacylglycerols (MGDG), digalactosyldiacylglycerols (DGDG), sulfoquinovosyldiacylglycerols (SQDG), and a porphyrin (Figure 5, Table 1) were reported from the *Sargassum* species.

Recently, crude extracts of 18 brown algae collected along the Brazilian coast were evaluated for their photoprotective capacity in response to abiotic factors and geographic distribution. The results highlighted seven main peaks in the absorption spectra of the algae crude extracts. In the UV-C (230–280 nm), UV-B (280–320 nm), and UV-A (320–400 nm) regions, absorbing bands were recorded between 240–249 nm, 250–280 nm, 305–347 nm, and 358–377, respectively. Additionally, in the photosynthetically active radiation region (PAR), between 400–700 nm, main bands were observed at 400–425 nm, 440–455 nm, 456–480 nm, and 650–670 nm. The highest photoprotective index values were found in species collected in the northeastern part of Brazil, which is influenced by the Brazilian current and by the proximity to the Equator. The results obtained also revealed that the water temperature and nitrate concentration were negatively correlated with UV indices. Characteristics of the absorption spectrum of phenolic compounds and carotenoids were observed in all analyzed samples [28]. 

The correlation of abiotic factors with antioxidant activity was also observed by Vasconcelos et al. (2019). The authors evaluated the antioxidant activity of *Sargassum furcatum* using different spectrophotometric methods (DPPH, ABTS, MCA, Folin-Ciocalteu, and FRAP). The results revealed that *S. furcatum* extracts presented higher antioxidant activity in the DPPH and ABTS assays, with EC_50_ values of 0.461 ± 0.006, and 0.266 ± 0.009 mg/mL, respectively [29].

Another species of the *Sargassum* genus, *S. vulgare,* was investigated for antioxidant activity and the inhibition of the human immunodeficiency virus reverse transcriptase (RT-HIV). Different crude extracts were obtained, including methanolic, aqueous, and hot aqueous. Additionally, the bioactivity potential was measured by comparing extracts prepared using algal samples collected during the dry and rainy seasons. The methanolic extract from *S. vulgare* collected in the dry season showed the highest *β*-carotene bleaching activities (EC_50_ = 18.22 ± 2.91 μg/mL), while the hot aqueous extracts showed the highest RT-HIV inhibition and antioxidant activities in the ABTS, FRAP, and Folin-Ciocalteu assays. Furthermore, *S. vulgare* extracts collected in the dry season showed anti-HIV activity with an IC_50_ 10.15 ± 1.77 µg/mL and IC_50_ = 22.41 ± 5.74 μg/mL for extracts obtained from the rainy season [30]. 

A study reported the antioxidant activity of fifteen seaweeds collected along the Brazilian coast. In this investigation, only beach-cast marine algae were collected. Among the investigated species, brown algae such as *Dictyopteris jolyana* and *Zonaria tournefortii* presented the highest antioxidant potential [31]. 

Another investigation evaluated the antioxidant, photoprotective, and cytotoxic properties of extracts from beach-cast seaweed species. The brown seaweeds presented the highest antioxidant potential. *Zonaria tournefortii* presented the highest absorbance in the UVA-UVB region. None of the extracts presented cytotoxic effects on human normal cell lines [32]. Beach-cast marine algae are usually not used for the prospection of molecules focused on natural product discovery since it is not possible to determine their place of origin or the degradation products that will be generated. However, they represent large unused biomass amounts with the potential for the development of new bioproducts.

The antioxidant capacity responses to abiotic influences allow for an understanding of the species’ tolerance and sensitivity, and could be employed as parameters in the cultivation of commercially important species. Thus, a work by Urrea-Victoria and coworkers (2022) evaluated the antioxidant capacity and the phenolic compounds of *Sargassum stenophyllum* under different temperatures. The results showed *S. stenophyllum*, and presented high values of the antioxidant index and a high content of phlorotannins at a temperature of 30 °C [33]. 

Besides the antioxidant activity, metabolites isolated from seaweeds have shown antiviral potential against human type 1 herpes, human immunodeficiency virus, and Dengue [34,35,36,37,38,39]. Due to the current epidemic scenario of arthropod-borne viruses such as Dengue, Zika, and Chikungunya in Brazil [40], it is critical to discover compounds that can be used as prototypes for the development of novel antiviral medications. 

A number of studies have addressed this public health issue and evaluated the antiviral potential of seaweed collected in Brazil. Seaweeds from the *Dictyota* genus have been targeted in a number of prospective studies due to produce structurally diverse secondary metabolites, as defensive mechanisms [41].

The metabolites of *Dictyota menstrualis* had their antiviral potential evaluated against the Zika virus (ZIKV), alone and in combination with ribavirin. Two fractions, named F-6 and FAc-2, were able to inhibit ZIKV replication by >74% in a dose-dependent manner. Fractions F-6 and Fac-2 displayed EC_50_ values of 0.81 and 2.80 μg/mL, respectively. To disclose the mechanism of action, these fractions were evaluated regarding their virucidal profile and inhibition of viral adsorption using Vero cell models. Fraction FAc-2 presented strong virucidal potential, and fraction F-6 inhibited viral adsorption. Associating the FAc-2 fraction with ribavirin at suboptimal doses produced a strong synergistic effect that completely inhibited viral replication. Using the ^1^H NMR technique, the authors were able to propose the major constituents in the fractions F-6 and FAc-2. The ^1^H NMR spectra of fraction F-6 suggested the presence of acetylated cycloxeniane (**1**,**2**) (Figure 4) as the major constituent. In fraction FAc-2, ^1^H NMR resonances indicated dichotomane diterpenes (**3**,**4**) (Figure 4) as the most abundant constituents [42].

*D. menstrualis* crude extracts, acetylated crude extracts, and fractions were evaluated regarding their antiviral potential against the Chikungunya virus. The acetylated crude extract referred to by the authors aims to preserve unstable metabolites during the purification steps and increase the yield of diterpenes by the acetylation of secondary hydroxyl groups. During the biological evaluation, the active extracts were further fractionated, whereas fractions F-5 and FAc-1 were able to inhibit viral replication by >96% at 20 µg/mL and presented EC_50_ values of 0.90 and 0.73 µg/mL, respectively. In the virucidal assay, F-5 was the most active, reducing Chikungunya virus infectivity by 88% at 20 µg/mL. The ^1^H NMR spectral data of F-5 revealed the acetylated cycloxeniane (**2**) (Figure 4) as a major metabolite. In FAc-1, a dichotomane diterpene pachydictyol A (**5**) was identified as the major constituent [43]. 

In a recent study, the diterpene dolastane (**6**) (Figure 4) obtained from *Canistrocarpus cervicornis* was tested for its ability to suppress Zika and Chikungunya virus replication. Dolastane (**6**) (Figure 4) inhibited Zika and Chikungunya at values of EC_50_ = 0.95 µM and 1.3 µM, respectively. The compound was reported to possess virucidal activity against Chinkungunya, inhibiting viral infectivity by 90% at a dosage of 10 µM. In addition, dolastane inhibited Zika virus infectivity by 64% at the same concentration [44]. 

Four different brown algae: *Canistrocarpus cervicornis*, *Dictyopteris jolyana*, *Dictyopteris plagiogramma*, and *Dicyota mertensii* were chemically profiled using molecular networking. In general, the results showed that brown algae of the genus *Dictyopteris* mainly produced C_11_-hydrocarbons, sesquiterpenes, and sulfur-containing compounds, while *Dictyota* and *Canistrocarpus* are reported to mainly contain diterpenes. A diterpene, pachydictol A (**5**) (Figure 4), was isolated from *Dictyota mertensii*. In addition, the compound dictyoxide (**7**) (Figure 4) was also annotated for this species. The compounds 4b-hydroxydictyodial (**8**) (Figure 4), 4b-acetoxydictyodial A (**9**) (Figure 4), 18,4-dihydroxydictyo-19-al A (**10**) (Figure 4), 18-acetoxy-4-hydroxydictyo-19-al (**11**) (Figure 4), and dictyol B (**12**) (Figure 4) were identified for *D. crenulata*, while dictyol B acetate (**13**) (Figure 4), and dictyotadiol (**14**) (Figure 4) were identified for *D. dichotoma* [45]. 

Atomaric acid (**15**) (Figure 4), a meroditerpene, isolated from the brown seaweed *Stypopodium zonale,* was evaluated against the parasite that causes Chagas disease, *Trypanosoma cruzi.* The results showed that this compound can be considered a hit, demonstrating its potent trypanocidal effect with an EC_50_ value of 2.4 ± 1.8 µg/mL (5.4 µM) and an SI of 16.8 [46].

Typically, marine organisms do not rely on external coats such as hair, feathers, and waxes for protection; some have cuticles, shells, and spines, but the majority of species survive by escape, hiding, and chemical defenses. However, fouling is one of the most persistent biological stressors faced by marine species, and any undefended surface gets overtaken by microorganisms via biofilm development and, later, by macro-foulers within days or weeks [47].

Currently, the discovery of environmentally friendly antifouling agents is the focus of marine biotechnologists. Since seaweeds need to modulate microbial colonization on their surface, they produce compounds capable of balancing this ecological relationship, which could be used in the development of new antifouling agents [48]. 

In this sense, a recent investigation evaluated the antifouling activity of the seaweed *Sargassum vulgare* collected along the coast of southeastern Brazil. The investigators found that fractions containing monogalactosyldiacylglycerols—MGDG (**16**) (Figure 5), digalactosyldiacylglycerols–DGDG (**17**) (Figure 5), and sulfoquinovosyldiacylglycerols—SQDG (**18**) (Figure 5) presented promising antifouling activity against marine bacteria and microalgae [49]. 

Pheophytin (Sp-1) (**19**) (Figure 5) isolated from *Sargassum polyceratium* was evaluated regarding its antibacterial and antibiotic-modifying activity in strains of *S. aureus* with efflux pumps. Compound **19** did not present significant antibacterial activity; however, in combination with erythromycin and norfloxacin, it changed the antibiotic activity, decreasing bacterial resistance by two to four times [50]. 

The schistosomicidal activity of crude extracts from eight species of brown seaweeds were investigated using different extraction methods, including different organic solvents. Among the evaluated species, *Dictyota mertensii* and *Dictyota ciliolate* presented the highest schistosomicidal activity. The *D. mertensii* extract generated by supercritical fluid induced 100% lethality after 48 h of exposure, and affected schistosoma reproduction, with 80% of pairs separated after 2 h of exposure and 100% after 24 h. The oviposition was totally inhibited. The *D. ciliolata*, schistosomicidal activity was influenced by the solvent used in the extraction. Supercritical fluid showed the highest lethality (100%) and also affected reproduction and oviposition, with 100% of pairs separated after treatment. Through a combination of GC-MS analysis and PLSDA, compounds such as 9-acetoxydichotoma-2,13-diene-16,17-dial (**20**), dictyol B (**13**) (Figure 4), dictyol C (**21**) (Figure 4), isopachydictyol A (**22**) (Figure 4), and dictyotadiol (**14**) (Figure 4) were putatively annotated as biomarkers of bioactivity [51].

#### 2.1.2. Chlorophyta

Caulerpin (**23**) (Figure 6), was isolated from the marine green alga *Caulerpa racemosa* and evaluated for anti-Chikungunya virus (CHIKV) activity. This compound presented 99% inhibition at concentrations of 20, 10, and 5 μM. The concentrations of 2.5, 1.25, and 0.65 μM showed 92%, 75%, and 47% inhibition, respectively. The EC_50_ value was 0.8 μM, and caulerpin (**23**) (Figure 6) presented a high selectivity index (SI) of 736.62, which suggests its selectivity for the virus [52].

The genotoxicity and osteogenic potential of sulfated polysaccharide (SP)-enriched samples extracted from other species of *Caulerpa* (*C. prolifera* and *C. sertulariodes*) were evaluated in the cytokinesis-block micronucleus assay and on human mesenchymal stem cells (MSCs), respectively. For *C. prolifera*, the subfraction at 5 µg/mL presented the most promising results, since it did not induce micronuclei, proving to be non-genotoxic. In addition, human MSCs were induced to differentiate into osteoblasts using low amounts of this fraction [53]. The chemical composition analyses of the SPs extracts obtained from *C. sertulariodes* revealed a high sugar and sulfate content. Human MSCs treated with seaweed SPs (1–10 μg/mL) did not alter 3-(4,5-methylthiazol-2-yl)-2,5-diphenyltetrazolium bromide reduction activity, and alkaline phosphatase activity was increased by approximately 30–40% when cells were treated with 5–10 μg/mL SPs [54]. Therefore, these results suggest the potential of SPs extracted from green seaweed for therapeutic applications in bone regeneration.

In general, SP molecules are analogous to mammalian glycosaminoglycans, and due to their origin, they are potentially safer for pharmacological applications. In particular, SPs extracted from seaweed have demonstrated many biological functions, such as antioxidant, anticoagulant, antitumor, and antiviral activities, among others [55,56,57]. In this regard, the SF-rich extracts obtained from seven green seaweeds (*C. cupressoides*, *C. sertulariodes*, *C. prolifera*, *C. racemosa*, *Codium isthmocladum*, *Udotea flabellum*, and *Ulva lactuca*) collected in northeastern Brazil were evaluated for their role as anticoagulants. Among these, *U*. *flabellum* extract was the most potent. Two sulfated homogalactans with anticoagulant activity, F-I (130 kDa) and F-II (75 kDa), were isolated from this extract through several bio-guided purification steps. Although F-II had a higher content of sulfate groups than F-I, the activities of both polysaccharides were similar across all assays. F-I and F-II also inhibited murine melanoma cells’ (B16-F10) adhesion, migration, and proliferation on a fibronectin-coated surface, but not on laminin or collagen I-coated surfaces. In conclusion, the antiproliferative activity of F-I and F-II was dependent on their degree of sulfation [58].

The ability to inhibit the RT-HIV enzyme and the antioxidant activity of green seaweeds has also been investigated. The methanolic extract of seaweed *Ulva fasciata* showed antioxidant activity in the *β*-carotene/linoleic acid assay, with an EC_50_ value of 33.41 ± 1.53 μg/mg, but no anti-HIV effect was observed [30]. The schistosomicidal activity of green algae has also been investigated. The species *Codium isthmocladum* presented weak activity by inducing 20% mortality in male worms; however, no effects were observed in females. *C. isthmocladum* extracts also affected reproduction with 100% separation of pairs. *Caulerpa sertularioides* and *C. racemosa* extracts significantly affected reproduction, caused 100% and 80% pair separation, respectively, and almost completely inhibited oviposition [51].

#### 2.1.3. Rhodophyta

The antidiarrheal activity of sulfated polysaccharides (SPs) extracted from the red seaweed *Gracilaria cervicornis* was evaluated in Swiss mice that were pre-treated with castor oil or cholera toxin. The *G. cervicornis* SPs reduced the total diarrheic stools excreted and the intestinal fluid accumulation by 61.87% and 53.54% in the SP doses of 1 and 3 mg/kg, respectively. In addition, it was able to increase the Na^+^/K^+^-ATPase activity in the small intestine in acute diarrhea and inhibit the toxin-GM1 binding. The SPs also decreased the secretion of fluid and chloride ions excreted in secretory diarrhea [59].

A sulfated polysaccharide (**24**) (Figure 7), was extracted from the seaweed *Digenea simplex,* and was evaluated regarding its anti-inflammatory effect on 2,4,6-trinitrobenzene sulfonic acid–induced colitis in rats. Compound **24** (Figure 7) is composed of *β*-d-galactose and 3,6-α-l-anhydrogalactose residues and was found to reduce wet weight and macroscopic and microscopic lesion scores. The myeloperoxidase activity, proinflammatory cytokines, malondialdehyde, and nitrate/nitrite levels were also reduced by compound **24**, which preserved glutathione consumption in the colon [60].

The lectin extracted from *Bryothamnion triquetrum* had its anti-inflammatory activity investigated in mice using several classical models of inflammation. The results showed that lectin inhibited paw edema induced by carrageenan/dextran and myeloperoxidase activity, in addition to decreasing cell migration and TNF-α and IL-1β production in peritonitis induced by carrageenan. Therefore, the lectin produced by *B. triquetrum* holds promise for evaluation in clinical trials [61]. 

The crude extracts (methanolic, aqueous, and hot aqueous) of the seaweed *Palisada flagellifera* from dry and rainy seasons had their antioxidant and anti-HIV activities evaluated. The results of this study showed that the methanolic extract presented an EC_50_ 24.85 ± 3.13 μg/mg in antioxidant power in a *β*-carotene/linoleic acid assay. Meanwhile, the best antiviral responses for *P. flagellifera* were for aqueous extracts with IC_50_ values of 277.60 ± 44.46 and 200.07 ± 57.14 μg/mL for the dry and rainy seasons, respectively [30]. Terpenes and halogenated terpenes were also reported for seaweeds from the *Laurencia* and *Plocamium* genera (Figure 8).

The metabolites obtained from the red seaweed *Plocamium brasiliense* were evaluated for their activity against the parasite *T. cruzi*. A monoterpene, 5-chloro-1-(*E*)-chlorovinyl-2,4-dibromo-1,5-dimethylcyclohexane (**25**) (Figure 8) and a mixture of halogenated monoterpenes had enhanced trypanocidal activity and showed IC_50_ values of 7.1 ± 2.1 and 4.9 ± 3.7 µg/mL, respectively, against *T. cruzi* intracellular amastigote forms. Although the samples showed low selectivity, these results identify monoterpenes as promising candidates against Chagas disease [46]. 

*Kappaphycus alvarezii* is farmed globally, mainly for carrageenan extraction. However, there is increasing interest regarding the biotechnological potential of its secondary metabolites. This fast-growing seaweed has harvest cycles of 100–120 days, and the cultivation of these species is an important industry in Indonesia, the Philippines, Malaysia, Zanzibar, Solomon Island, and Micronesia [62]. In Brazil, the cultivation of this seaweed is incipient, and research is encouraged with the aim of adding value to the bioproducts of this species. Recently, the antioxidant properties of different strains of *K. alvarezii* Brazil were investigated. The antioxidant activity of methanolic and aqueous extracts of brown, green, red, and G11 strains were evaluated by Folin–Ciocalteu reducing potential, and the DPPH, ABTS, and FRAP assay methods. The results indicated that the green strain presented the highest antioxidant index when compared to the other strains, suggesting its prioritization as a source of molecules with antioxidant properties [63]. Another investigation evaluated the seasonal variation in the nutritional and antioxidant potential of *K. alvarezii.* The results highlighted that the green strain presented the highest mineral content. In contrast, a higher carbohydrate level was observed in the brown strain. The brown strain also presented the highest antioxidant potential of *K. alvarezii* for samples collected during the spring [64]. 

Allelopathy is an ecological strategy used by plants to promote interspecific and intraspecific competitive ability through the production of metabolites. It also influences community organization by modulating patterns of the spatial distribution of species [65]. Despite this process being well documented in plants, little is known about allelopathy in seaweeds. In a recent investigation, crude extracts, and the pure compounds elatol (**26**) (Figure 8) and obtusol (**27**) (Figure 8) from two distinct populations of *Laurencia dendroidea* were investigated to evaluate the effect of autotoxicity through auto- and crossed experiments by chlorophyll fluorescence imaging to measure the inhibition of photosynthesis as a variable response. The results revealed that seaweed from Azeda Beach were inhibited by extracts (IC_50_ = 219 μg/mL and IC_50_ = 87 μg/mL). The extract and elatol also inhibited the individuals from Forno Beach (IC_50_ = 194 μg/mL for the extract and IC_50_ = 277 μg/mL for elatol). The extract of *L. dendroidea* from Forno Beach and the isolated obtusol (**27**) (Figure 8) showed no inhibitory effect in individuals of both populations. This finding highlights the potential of seaweed secondary metabolites for intra- and inter-populational interactions and for structuring seaweed populations [66].

Recently, 25 species of Rhodophyta were screened for their anthelmintic activity, of which five exhibited schistosomicide activity towards *S. mansoni*. Among them, the different extracts (hexane, chloroform, and methanolic) of four species from the *Laurencia* complex were bioactive, including *L. aldingensis*, *L. catarinensis*, *L. dendroidea*, and *Laurenciella* sp. The *L. aldingensis* hexane extract induced 100% mortality in worms, with selective activity towards female worms. The reproduction was also affected, and worm pairing showed 80% separation by hexane extract. The chloroform extract caused the 100% separation of worm pairs. The *L. aldingensis* hexane and chloroform extracts also inhibited oviposition by 100%. The *L. dendroidea* hexane and chloroform extracts induced 100% mortality in worms. The effects of reproduction were also solvent-dependent, with pairs separation ranging from 60% to 100% for methanolic, hexane, and chloroform extracts, respectively. The hexane and chloroform extracts also totally inhibited oviposition. The chloroform extract of *L. catarinensis* was not active against worms, but was able to induce 100% pair separation and oviposition. The chloroform extract of *Laurenciella* sp. presented schistosomicide activity, with 100% mortality. Reproduction was inhibited by all extracts, with 100% separation of the worm pairs exposed to the hexane extract. The oviposition was highly inhibited by the hexane and chloroform extracts. The *L. catarinensis* chloroform extract induced 100% pair separation and oviposition. Through the combination of GC-MS analysis and PLS-DA, discriminant metabolites in the active extracts were targeted and annotated as a silphiperfolanol derivative and silphiperfolan-7β-ol (**28**) (Figure 8) [51].

The halogenated sesquiterpenes, (−)-elatol (**26**), (+)-obtusol (**27**), and (−)-rogiolol (**29**) (Figure 8) were screened against *S. mansoni* and *B. glabrata*. The *S. mansoni* adult worms were exposed to compounds **25**–**27** for 96 h. Rogiolol (**29**) (Figure 8) induced 100% by the end of the exposure time. Additionally, compound **27** (Figure 8) also affected the worms’ reproduction, inducing 100% couple separation after 24 h and the total inhibition of oviposition. (−)-Elatol (**26**) (Figure 8) affected reproduction, with 100% separation of worm couples and the total inhibition of oviposition. *S. mansoni* cercariae larvae were exposed to 25 µg/mL of each compound, and induced 100% of mortality after 5 min of exposure. These compounds also induced 100% mortality in *B. glabrata* embryos at 25 µg/mL. At this level, all the analogs induced 100% mortality of the embryos both at the blastulae and veliger stages [67].

The antioxidant potential and the accumulation of mycosporine-like amino acids (MAAs) in response to temperature changes have been investigated in the seaweed *Pyropia spiralis* [33]. The reported results showed that extracts of *P. spiralis* cultivated at 15 °C exhibited the highest antioxidant potential and a higher content of the MAAs porphyra-334, shinorine, and palythine. Since the antioxidant potential decreases as temperature increases, the investigators hypothesized that the accumulation of MAAs may be a protective mechanism against freezing [33].

Three algae of the genus *Laurencia, L. catarinensis, L. dendroidea*, and *L. intricata* had their metabolic profiles evaluated using a molecular networking approach. Algae belonging to this genus are rich in halogenated secondary metabolites, mainly sesquiterpenes and C_15_-acetogenins. Three chamigrane-type sesquiterpenes, prepaciphenol epoxide (**30**), johnstonol (**31**) (Figure 8), and pacifenol (**32**) (Figure 8) were isolated from *L. catarinensis*, and others were annotated as laucapyranoids B (**33**) (Figure 8) and C (**34**) (Figure 8), laurendecumallenes A (**35**) and B (**36**) (Figure 8), laureepoxide (**37**) (Figure 8), (3*E*)-elatenyne (**38**) (Figure 8), elatenyne (**39**) (Figure 8), kumausallene (**40**) (Figure 8), and laurobtusin (**41**) (Figure 8). No compound could be identified in the extracts of *L. dendroidea* and *L. intricata* [45].

An overview of the reported compounds for Ochrophyta, Chlorophyta, and Rhodophyta, as well as the investigated bioactivities and the solvents used for extraction procedures is provided in Table 1.

**Table 1 molecules-28-04285-t001:** Seaweeds from the Brazilian coast (studies published between 2018–2022): Biological activities, isolated compounds, and other characteristics.

Reference	Seaweed	Class	Site of Collection	Biological Activity	Isolated Compounds
Schmitz et al., 2018 [28]	*Asteronema breviarticulatum* *Bachelotia antillarum* *Canistrocarpus cervicornis* *Chnoospora minima* *Colpomenia sinuosa* *Dictyopteris delicatula* *Dictyopteris justii* *Dictyopteris plagiogramma* *Dictyota ciliolate* *Dictyota mertensii* *Levringia brasiliensis* *Lobophora variegate* *Padina boergesenii* *Padina gymnospora* *Padina sanctae-crucis* *Rosenvingia* sp. *Sargassum* spp. *Spatoglossum schroederii*	Ochrophyta	Twenty-three sites distributed along the coastline of Brazil	Photoprotective (UV_C_i, UV_B_ + _A_i and PARi)	Dichloromethane:methanol extract (1:1)
Vasconcelos et al., 2019 [29]	*Sargassum furcatum*	Ochrophyta	Enseada dos Corais Beach—Pernambuco (8°18′44.7″ S; 34°56′49.8″ W)	Antioxidant (DPPH, ABTS, Metal chelator activity, Folin-Ciocalteu, and FRAP)	Dichloromethane:methanol extract (2:1)
	*Bryothamnion triquetrum* *Osmundaria obtusiloba*	Rhodophyta			
Santos et al., 2019 [30]	*Sargassum vulgare*	Ochrophyta	Morro de Pernambuco beach—Bahia (14°48′21.6″ S, 39°01′25.6″ W)	Antioxidant (DPPH, *β*-carotene/linoleic acid system, ABTS, FRAP, metal chelating, and Folin-Ciocalte) and anti-HIV	Methanolic, aqueous, and hot aqueous extracts
	*Palisada flagellifera*	Rhodophyta			
	*Ulva fasciata*	Chlorophyta			
Harb et al., 2021 [31]	*Dictyopteris jolyana* *Dictyopteris polypodioides* *Zonaria tournefortii*	Ochrophyta	Pontal Beach-Espirito Santo (20°58′22.5″ S; 40°48′38.6″ W) Coqueirinho Beach-Paraiba (07°17′58″ S; 34°47′54″ W) Ponta do Cabo Branco Beach—Paraiba (07°08′43.6″ S; 34°48′20.7″ W)	Antioxidant (DPPH, ABTS, Metal chelator activity, and FRAP)	Methanolic extracts
	*Agardhiella ramosissima* *Alsidium seaforthii* *Alsidium triquetrum* *Botryocladia occidentalis* *Gracilaria domingensis* *Osmundaria obtusiloba* *Spyridia clavata*	Rhodophyta	Itaoca Beach—Espirito Santo (20°54′18.0″ S; 40°46′42.3″ W) Piúma Beach—Espirito Santo (20°50′31.5″ S; 40°43′46.0″ W) Emboaca Beach—Ceara (3°12′23.5″ S; 39°18′37.1” W) Ponta do Cabo Branco Beach—Paraiba (7°08′43.6″ S; 34°48′20.7″ W)		
	*Codium isthmocladum*	Chlorophyta	Itaoca Beach—Espirito Santo (20°54′18.0″ S; 40°46′42.3″ W)		
Harb et al., 2022 [32]	*Dictyopteris jolyana* *Dictyopteris polypodioides* *Zonaria tournefortii*	Ochrophyta	Pontal Beach—Espirito Santo (20°58′22.5″ S; 40°48′38.6″ W) Coqueirinho Beach—Paraiba (07°17′58″S; 34°47′54″ W) Ponta do Cabo Banco Beach—Paraiba (07°08′43.6″ S; 34°48′20.7″ W) Pontal—Espirito Santo (20°58′22.5″ S; 40°48′38.6″ W)	Antioxidant (ABTS and Folin-Ciocalteu), photoprotection (ESAR and EPI), and cytotoxicity (HCT116, HL60, HGF1, and HaCat)	Alkaline aqueous and hydroethanolic extracts
	*Alsidium seaforthii* *Osmundaria obtusiloba*	Rhodophyta	Piuma—Espirito Santo (20°50′31.5″ S; 40°43′46.0″ W) Ponta do Cabo Branco—Paraiba (7°08′43.6″ S; 4°48′20.7″ W)		
	*Codium isthmocladum*	Chlorophyta	Itoca—Espirito Santo (20°54′18.0″ S; 40°46′42.3″ W)		
Urrea-Victoria et al., 2022 [33]	*Pyropia spiralis*	Rhodophyta	Cibratel Beach—Sao Paulo (24°13′31” S and 46°51′7” W)	Antioxidant (ABTS, DPPH, FRAP, metal chelating, Folin-Ciocalteau, and TAC)	Methanolic extracts
	*Sargassum stenophyllum*	Ochrophyta			
Cirne-Santos et al., 2019 [42]	*Dictyota menstrualis*	Ochrophyta	Enseada do Forno—Rio de Janeiro (22°44′00″ S; 41°57′25″ W)	Antiviral (Zika Virus)	5-acetoxy-1,6-cycloxenia-2,13-diene-16,17-dial (1), 5-hydroxy-1,6-cycloxenia-2,13-diene-16,17-dial (2), 6-hydroxy-dichotoma-2,13-diene-16,17-dial (3), 6-acetoxydichotoma-2,13-diene-16,17-dial (4)
Cirne-Santos et al., 2020 [43]	*Dictyota menstrualis*	Ochrophyta	Enseada do Forno—Rio de Janeiro (22°44′00″ S; 41°57′25″ W)	Antiviral (Chikungunya Virus)	Pachydictyol A (5)
Cirne-Santos et al., 2020 [44]	*Canistrocarpus cervicornis*	Ochrophyta	Praia do Velho—Rio de Janeiro (23°01′ S, 44°00′ W)	Antiviral (Zika and Chikungunya Virus)	Dolastane (6)
Philippus et al., 2018 [45]	*Dictyopteris plagiogramma Dictyopteris jolyana* *Dictyota mertensii* *Canistrocarpus cervicornis*	Ochrophyta	Rocas Atoll—Rio Grande do Norte Fernando de Noronha—Pernambuco Trindade Island—Espirito Santo Sao Pedro and Sao Paulo Archipelago—Pernambuco	-	Pachydictol A (5) Compounds annotated (LC-MS): Dictyoxide (7), 4*β*-hydroxydictyodial (8), 4*β*-acetoxydictyodial A (9), 18,4-dihydroxydictyo-19-al A (10), 18-acetoxy-4-hydroxydictyo-19-al (11), and dictyol B (12), dictyol B acetate (13), and dictyotadiol (14)
	*Laurencia catarinensis* *Laurencia dendroidea* *Laurencia intricata*	Rhodophyta	Xavier, Arvoredo Island, and Bombinhas—Santa Catarina Rocas Atoll—Rio Grande do Norte		Prepacifenol epoxide (30), johnstonol (31), and pacifenol (32) Compounds annotated (LC-MS): Laucapyranoids B (33) and C (34), laurendecumallenes A (35) and B (36), laureepoxide (37), (3*E*)-elatenyne (38), elatenyne (39), kumausallene (40), and laurobtusin (41)
Teixeira et al., 2019 [46]	*Plocamium brasiliense*	Rhodophyta	Enseada do Forno—Rio de Janeiro (22°44′49″ S and 41°52′54″ W)	Antiparasitic (*Trypanosoma cruzi*)	5-chloro-1-(*E*)-chlorovinyl-2,4-dibromo-1,5-dimethylcyclohexane (25)
	*Stypopodium zonale*	Ochrophyta			Atomaric acid (15)
Plouguerné et al., 2020 [49]	*Sargassum vulgare*	Ochrophyta	Itacuruca Island—Rio de Janeiro (22°56′ S, 43°52′ W)	Antibacterial (*Pseudoalteromonas elyakovii*, *Halomonas marina*, *Shewanella putrefaciens*, and *Polaribacter irgensii*) and Anti-microalgal (Chlorarachnion reptans, *Pleurochrysis roscoffensis*, *Exanthemachrysis gayraliae*, *Cylindrotheca closterium*, and *Navicula jeffreyi*)	Monogalactosyldiacylglycerols—MGDG (16), digalactosyldiacylglycerols—DGDG (17), and sulfoquinovosyldiacylglycerols—SQDG (18)
Menezes-Silva et al., 2020 [50]	*Sargassum polyceratium*	Ochrophyta	Joao Pessoa—Paraiba	Antibacterial (*Staphylococcus aureus*)	13^2^-hydroxy-(13^2^-R)-pheophytin-*a* (19)
Stein et al., 2021 [51]	*Amphiroa fragilissima* *Bostrychia tennela* *Botryocladia occidentalis* *Bryothamnion seaforthii* *Ceratodictyon variabile* *Cryptonenia crenulata* *Cryptonenia seminervis* *Dichotomaria marginata* *Gracilaria* cf. *intermedia* *Gracilaria domingensis* *Hypnea nigrescens* *Jania rubens* *Laurencia aldingensis* *Laurencia catarinensis* *Laurencia dendroidea* *Octhodes secundiramea* *Palisada perforata* *Palisada flagellifera* *Porphyra spiralis* *Pterocladiella capillacea* *Solieria filiformis* *Spyridia aculeata* *Tricleocarpa cylindrica* *Vidalia obtusiloba*	Rhodophyta	Intertidal zone—Espirito Santo	Antiparasitic (*Schistosoma mansoni*) and molluscicidal (*Biomphalaria glabrata*)	Dichloromethane, chloroform, hexane, and methanol extracts Compounds annotated (CG-MS): dictyol B (13), dictyotadiol (14), 9-acetoxydichotoma-2,13-diene-16,17-dial (20), dictyol C (21), isopachydictyol A (22), Silphiperfolan-7β-ol (28)
	*Canistrocarpus cervicornis* *Colpomenia sinuosa* *Dictyota ciliolata* *Dictyota mertensii* *Padina tetrastomatica* *Sargassum vulgare* *Zonaria tournefortii*	Ochrophyta			
	*Caulerpa cupressoides* *Caulerpa racemosa* *Caulerpa sertularioides* *Codium isthmocladum*	Chlorophyta			
Esteves et al., 2019 [52]	*Caulerpa racemosa*	Chlorophyta	Archipelago of Sao Pedro and Sao Paulo Pernambuco (00°55′ S, 29°21′ W)	Antiviral (Chikungunya Virus)	Caulerpin (23)
Chaves-Filho et al., 2018 [53]	*Caulerpa prolifera*	Chlorophyta	Natal—Rio Grande do Norte	Genotoxicity and osteogenic	Sulfated polysaccharide (SP)-enriched extract
Chaves-Filho et al., 2019 [54]	*Caulerpa sertularioides*	Chlorophyta	Coast Rio Grande do Norte	Osteogenic	Sulfated polysaccharide (SP)-enriched extract
Marques et al., 2019 [58]	*Caulerpa cupressoides* *Caulerpa sertulariodes* *Caulerpa prolifera* *Caulerpa racemosa* *Codium isthmocladum* *Udotea flabellum* *Ulva lactuca*	Chlorophyta	Buzios Beach—Rio Grande do Norte (05°58′23″ S, 35°04′97″ W) Rio do Fogo Beach—Rio Grande do Norte (05°16′22″ S, 35°22′58″ W) Camapum beach—Rio Grande do Norte (05°06′54″ S, 36°38′02″ W)	Anti-Thrombin, Anti-Adhesive, Anti-Migratory, and Anti-Proliferative	Sulfated polysaccharide (SP)-enriched extract
	*Lobophora variegata* *Sargassum vulgare* *Padina gymnospora* *Sargassum felipendula* *Spatglossum schröederi* *Dictyota mertensii* *Dictyopteris delicatula* *Dictyota menstrualis* *Canistrocarpus cervicornis* *Dictyopteris justii* *Dictyota ciliolata*	Ochrophyta			
	*Gracilaria birdiae* *Gracilaria caudata* *Amansia. multifida* *Achantophora especifera*	Rhodophyta			
Bezerra et al., 2018 [59]	*Gracilaria cervicornis*	Rhodophyta	Flecheiras Beach—Ceara	Antidiarrheal	Sulfated polysaccharide (SP)-enriched extract
Monturil et al., 2020 [60]	*Digenea simplex*	Rhodophyta	Flexeiras Beach—Ceara (03°13′25″ S and 39°16′65″ W)	Anti-inflammatory	Sulfated polysaccharide (24)
Fontenelle et al., 2018 [61]	*Bryothamnion triquetrum*	Rhodophyta	Flecheiras Beach—Ceara	Anti-inflammatory	Lectin
Araújo et al., 2020 [63]	*Kappaphycus alvarezii*	Rhodophyta	Fisheries Institute—Sao Paulo (23°27.134′ S, 45°02.817′ W)	Antioxidant (Folin–Ciocalteu, DPPH, ABTS and FRAP	Methanolic and aqueous extracts
Sudatti et al., 2020 [66]	*Laurencia dendroidea*	Rhodophyta	Forno Beach (22°58′0003.3″ S, 42°00′56.2″ W) and Azeda Beach—Rio de Janeiro (22°44′33.6″ S, 41°52′055.6″ W)	Allelopathy and autotoxicity	(+)-elatol (25) and obtusol (27)
Dos Santos et al., 2022 [67]	*Laurencia dendroidea*	Rhodophyta	Ubu and Castelhanos beach (20°48′6″ S, 40°35′37″ W) and Praia Brava—Sao Paulo (24°37′47″ S, 45°12′6″ W)	Antiparasitic (*Schistosoma mansoni*) and molluscicidal (*Biomphalaria glabrata*)	(+)-obtusol (27), (−)-elatol (26), and (−)-rogiolol (29)

#### 2.1.4. Seaweed Essential Elements

From 1900, with the expansion in food manufacturing around the world, some components (hydrocolloids) present in seaweed, such as alginate, carrageenan, and agar, were used in the nutraceutical, pharmaceutical, and biotechnological industries due to their gelling properties in foods [20,68]. However, less than 1% the of the global production of seaweed is currently used for health purposes, despite the nutritional benefits of its substances [69,70]. Approximately 66% of known algae (in general) species are used as food, and Asian countries are the largest consumers, and use them in various culinary forms. Thus, some species of seaweed can be consumed due to their nutritional benefits, and are marketed in natura, in flakes, flour, powder, or incorporated into other food products [21,70].

Seaweeds are vital for ocean life, providing shelter, food, and intermediating ecological interactions. Since antiquity, seaweeds have been consumed as food (in fresh salads, soups, garnishes, soups, sushi, temake, etc.), especially in Asia, where species such as Nori (purple seaweed) stands out in cooking [71]. Currently, they have been incorporated into vegan and vegetarian diets, in the production of food supplements, in the pharmaceutical and cosmetics industries, in biodiesel production, in animal feed, and in wastewater treatment [72,73], as they are rich in vitamins, proteins, amino acids, and macro- (calcium, phosphorus, magnesium, potassium, etc.) and micro-elements (copper, iron, manganese, zinc, among others) [74,75,76]. 

Essential elements such as copper, manganese, iron, zinc, calcium, phosphorus, and magnesium are, in adequate concentrations, essential for the biological and biochemical functioning of several living beings [75]. For example, copper participates in various physiological processes, as it acts as a cofactor for numerous proteins, such as superoxide dismutase and ceruloplasmin. Its redox flexibility allows it to act as a vital mechanism in cellular respiration due to cytochrome C oxidase being an important copper-binding protein [77,78]. The presence of iron is indispensable for the correct development of numerous physiological functions in organisms (including marine ones). This element participates in several processes and functions, including oxygen metabolism (Krebbs cycle), electron transport (cytochromes, ferredoxins), the catalytic centers of enzymes, and the co-enzymes of different types (peroxidases, catalases, purple acid phosphatases). Iron deficiency can cause iron deficiency anemia and trigger symptoms such as fatigue, weakness, and shortness of breath, among others [79,80,81]. Manganese is an essential element involved in several enzyme systems that interact with or are dependent on it for catalytic regulatory functions. It is responsible for the correct metabolization of free radicals in the mitochondria, in addition to its participation in the production of glucose and the regulation of the immune system [82]. Similarly, zinc is essential to life, acting structurally and/or catalytically on more than 300 proteins. Zn occurs in nature in the form of oxides, sulfates, nitrates, and carbonates, and, due to its ability to form different geometric coordination, this element is involved in immune system defense, reproductive health, neurobehavioral activity, as well as in the synthesis and degradation of proteins, carbohydrates, and lipids. Zinc deficiency can lead to weight loss, delayed growth, wound healing and other conditions [83,84,85].

The nutritional benefits conferred by seaweed are important for human food security, but these data are scarce and do not assess the potential adverse effects. To expand these studies and investigate the effects (beneficial or toxic) of these chemical elements, it is necessary to develop analytical methodologies for the sample preparation (wet digestion, using inorganic acids, and oxidizing agents) of seaweed;, as well as bioaccessibility tests (simulating the physiological conditions of the organism during digestion, considering three areas of the digestive system: mouth, stomach, and intestine) and bioavailability (the evaluation of permeability and absorption) in the gastrointestinal tract [86]. In the literature, studies have investigated the bioaccessibility of phenolic compounds and carotenoids in algae [87,88,89,90,91,92,93,94], but few have evaluated essential and potentially toxic elements [95,96,97].

There are several types of in vitro digestion methods used for foods that aim to determine the bioaccessible fraction of essential and non-essential elements, phenolic compounds, and mycotoxins, among others. Some of these digestion models were standardized and are widely used to study the real nutritional value of foods, indicating that the total concentration does not provide enough information to determine the amounts for a daily diet [98,99]. Thus, identifying the real amount of nutrients or other chemical species (antinutrients, potentially toxic elements, etc.) that seaweeds can provide is important, given their high potential for the production of supplements and/or food.

#### 2.1.5. Patents Registered for Seaweed-Derived Products in Brazil

To provide an overview of the current innovation status of seaweed and coral-derived products in Brazil, we analyzed the patents deposited in the last five years. In the search conducted in the Espacenet database, no patent documents involving algae and corals from the Brazilian coast were found. In contrast, a search in the INPI returned 32 patents that described the biotechnological application of seaweeds from the Brazilian coast. No invention was registered for corals in the analyzed period. The results are summarized in Table 2, which describes the INPI identification number of the patent letter application, the title, and a summary with the main information of the invention. Additionally, Figure 9A,B shows the country responsible for the deposit and the proposed use or method registered in the patents. 

It is possible to note a lack of patents with the low number of deposits with the theme “algae and corals from the Brazilian coast with biotechnological use”. In the search on Espacenet (the sample was null) and at the INPI (the sample consisted of 32 documents), one should note that Brazil has a biotechnological potential added to its biodiversity, which should be used to guide the implementation plan of the country’s bioeconomy. However, Brazil has been recording a significant drop in patenting, as in 2013 there were 34,000 registrations at the INPI, and in 2020 this number was approximately 27,000. Between 2018 and 2022, there was a variation in the number of patents filed. This may be related to the scenario of the new Coronavirus pandemic, or even, to the registry process, which is lengthy and bureaucratic.

The sample of patents in the present study came from the INPI, a Brazilian agency which is administered by a body called the Center for the Dissemination of Technological Information (CEDIN), and which provides generic information such as catalog classification data from the International Patent Office and a summary of the document sought. However, out of the 32 patents analyzed, only 10 were deposited by Brazilian institutions or legal persons; the remaining 22 patents were deposited in Brazil, but the ownership of the rights belonged to institutions from different parts of the world. Additionally, it is worth remembering that the only year in which there was no registration of patents in which the owner was from Brazil was in 2022, which may change since there is a possibility that these documents are still in the concession process. The development of patents can occur through a legal entity or company, by an individual, via a teaching or research institution, or through a partnership between these institutions. The largest patent applicants in the years analyzed were private companies, with 22 applications (two in collaboration with research institutions); followed by teaching and research institutions with nine applications, and autonomous people, with two applications.

It is necessary to encourage the production of patents, particularly for research institutions, as it is a process that stimulates research that can enable the development of new technologies. This insertion can be made possible through partnerships, as was the case in some of the analyzed patents between sectors, and can be very favorable (even generating income) for scientific production.

In the period ranging from 2018 to 2022, there were twelve patents registered for seaweed’s use for agricultural purposes in the INPI (Table 2). Applications of the inventions included the use of seaweeds in fertilizers [100,101,102,103,104,105,106], in animal feeding [107,108,109], to improve plant productivity, and in damage repair [110,111]. 

Numerous studies have addressed the use of seaweed-based biostimulants, which can improve crop productivity by conferring resistance to abiotic stresses (drought, salinity), and soil degradation [112]. Seaweeds have proven to be an eco-friendly alternative to synthetic inputs, such as fertilizers or pesticides. Additionally, their extracts can reduce the resistance of pathogens, promote plant fertilization, increase productivity, improve plant health, and improve final product quality [113]. By feeding seaweed to cattle, they can boost productivity and even serve as a tool to reduce greenhouse gas emissions, which adheres to the principles of sustainable development [114]. 

We found patents registering innovations using seaweeds to promote human nutrition and health [115,116,117], describing antiparasitics and with regard to the treatment of immunosuppression [118,119,120], and applications in photodynamic therapy [121]. Patent BR 11 2020 025226 3 refers to the antiparasitic activity of seaweeds such as *Saccharina latissimi*, *Laminaria digitate*, *Ascophyllum nodosum*, and *Laminaria hyperborean* for inhibiting, reducing, and/or suppressing parasites [118]. Patent BR 10 2019 018113 3 also registered the use of marine algae derivatives with leishmanicidal activities. The invention proposes the use of *Prasiola crispa* fractions with a highly effective leishmanicidal effect, which provides a new non-toxic alternative for the treatment of this neglected disease [119]. Patent BR 10 2021 014827 6 refers to the process of obtaining stabilized polymeric nanoparticles of R-Phycoerythrin by encapsulation in a phycobiliprotein extracted and purified from red macroalgae in a polymeric matrix using the double emulsification method by solvent evaporation. The nanoparticle produced has greater spectroscopic stability (absorbance and fluorescence), in addition to greater permeability and retention in skin cancer cells, showing a potential for use in the diagnosis and treatment of the disease [121].

Patents were registered for miscellaneous medical devices such as biodegradable and antimicrobial bandages [122], and a dilutor for cryoprotection [123]. Patent BR 10 2019 027698 3 is an invention characterized by the use of algae of the genera *Hypnea* and *Dictyopteris* for the preparation of an extender (cryopreservative medium) for cryoprotection and the freezing of gametes. It was noted that in female gametes there was an improvement in viability and membrane integrity. In male gametes, the extender based on seaweed proved to be efficient in preserving sperm motility, membrane integrity, and functionality, which may increase the possibility of the successful fertilization of that gamete [123]. 

Patents regarding the obtaining of secondary metabolites were also recorded [124,125]. Seaweed’s secondary metabolites are commonly used in the cosmetic industry [126], and patents reporting its applications were also deposited [127,128]. 

Apart from agricultural, nutritional, pharmaceutical, and cosmetic applications, a number of miscellaneous applications were registered, including innovations such as biodegradable food packaging [129,130], the production of bio-hydrogen, and bioethanol using beach cast *Sargassum* in anaerobic reactors [131] and elastomer compositions [132]. Patent BR 10 2018 013380 2 registered a biodegradable film (biofilm), based on carrageenan to serve as food packaging [130]. Patent BR 10 2018 072892 0 provides the food and biodegradable plastic industries with an innovative process for the production of edible, recyclable, disposable, and biodegradable bio-cups and bio-utensils using agar-agar extracted from red seaweeds for the total or partial replacement of disposable and biodegradable utensils [129]. Seaweeds offer an alternative for packaging materials. Due to their rich polysaccharide content, they can be used as a raw material for developing sustainable packaging options. When combined with biodegradable polymers, seaweed-based packaging provides an environmentally friendly alternative to traditional packaging materials [133]. In addition, there were patents regarding new methods for the cultivation and maintenance of algae cultures [134,135,136]. Table 2 shows the summarized data of the patents deposited in Brazil in the last 5 years.

Brazil still does not have a strong patent culture in terms of an instrument of innovation but only as a source of market protection, which is regrettable. Thus, it is necessary to transform patents into something more practical for society, as Brazilian contractual and business procedures are still very complex, and these must be simplified so that patents become a more accessible and popular source of research.

**Table 2 molecules-28-04285-t002:** Summarized data of the patents deposited in Brazil in the last 5 years.

INPI Number/Reference	Seaweed	Patent Title	Summary	Depositor’s Country of Origin
BR 11 2019 015051 0 [136]	Not specified	System and method for growing algae	Aspects of the invention are directed to a system and method for homogenizing an algae cultivation vessel. The method may include controlling at least one first homogenizer to deliver a first fluid into the vessel at a first operating flow rate; and controlling at least one-second homogenizer for dispensing a second fluid into the vessel at a second operating flow rate. The first flow rate of operation can be adapted to allow the mixing of the algae in the culture vessel, and the second flow rate of operation can be adapted to allow the assimilation of materials into a liquid within the culture vessel.	United States of America (USA)
BR 11 2020 015303 6 [100]	*Ascophyllum nodosum*, *Laminaria* sp., *Fucus* sp., *Macrocystis pyrifera*, *Ecklonia maxima* or *Durvillea* sp.	Bioreactor and seaweed culture method	The present invention relates to a bioreactor, which includes a first compartment designed to retain seaweed sporophytes, a second compartment in fluid communication with the first compartment, which includes one or more seating surfaces, and a first porous barrier between the first and first compartments. The second compartment allows seaweed spores to pass from the first to the second compartment, and to systems comprising the bioreactor. Also provided herein are methods of culturing seaweed, for example by using a bioreactor.	United States of America (USA)
BR 11 2019 017483 4 [117]	Not specified	Oil/fat compositions containing fine food particles, methods for producing said compositions, and methods for increasing the water absorption index, increasing an opacity value, enhancing the extent of flavor, enhancing the sensation of swallowing, and to enhance the stability, toimprove the smoothness, and to improve an initial taste of said compositions	It has a composition that allows various vegetables, fruits, seaweeds, and the like to be stably present in the composition and has a wide utility in that the composition can be used in various applications. An oil/fat composition containing fine food particles comprising fine particles of at least one food selected from the group consisting of a vegetable, a fruit, and a seaweed; and an oil/fat, and having: (1) a food particle content of 2% by mass or more and 98% by mass or less, (2) a total oil/fat ratio of 10% by mass or more and 98% by mass or less, (3) a modal diameter in an ultrasonicated state of 0.3 µm or more and 200 µm or less, and (4) a water content of less than 20% by mass.	Japan (JP)
BR 11 2019 019497 5 [115]	Not specified	Innovative process for extracting protein from plant or algal matter	This is a method for separating proteins from plant materials or algae. The method comprises mixing the protein-containing material with a solvent, preferably water; extracting the material containing protein at pH > 7; and acidifying the mixture, thereby precipitating proteins and fibers together. In some embodiments of the invention, the separation further involves decanting the mixture to recover a protein/fiber solid; adding water to the protein/fiber solid; adding a predetermined amount of base to the protein/fiber/water system, thereby precipitating the fiber, separating the fiber from the protein in a decanter, and drying the protein solution. In other embodiments, the protein/fiber solid is processed directly, for example by passing it through an extruder. The use of fiber as a carrier for the protein makes the inventive method more efficient than methods known in the art; the inventive method does not require the use of a clarifying centrifuge.	United States of America (USA)
BR 11 2019 023615 5 [101]	*Phymatolithon*, *Lithothamnium*, *Ascophyllum*.	Water-disintegratable granular composition, process for preparing the water-disintegratable granular composition, method of fortifying crops and plants, use of the water-disintegratable granular composition, and method of improving plant health	The invention relates to a water-disintegratable granular composition, wherein the granules include, at least, one culture/algae nutrient, or an active water-insoluble pesticide.	India (IN)
BR 10 2018 013380 2 [130]	Not specified	Antimicrobial bioactive film based on carrageenan and olive leaf extract	The product is a biodegradable film or biofilm that is produced based on carrageenan and enhanced with olive leaf extract for the purpose of food packaging. Carrageenan is a biopolymer extracted from red algae with the potential to form good films. Already, the olive leaf extract can be considered a plant antimicrobial compound with a possible application as a food preservative. In this context, the proposed biodegradable film is an active packaging alternative, as in addition to protecting the food against undesirable external actions, it interacts with it for its conservation, since an antimicrobial auxiliary component is added to its constitution, which improves the performance of the packaging system.	Brazil (BR)
BR 11 2020 000675 0 [102]	Green algae, red algae, golden algae, brown algae, golden-brown algae, blue algae, blue-green algae.	New fortification, crop nutrition, and crop protection composition	The invention relates to an algae composition in the form of an aqueous suspension. More particularly, the invention relates to the aqueous suspension composition including one or more algae selected from green algae, red algae, golden algae, brown algae, golden-brown algae, blue algae, blue-green algae, or aspecies thereof in the range of 0.1% to 65% by weight with one or more surfactants in the range of 0.1% to 50% by weight; with one or more structuring agents in the range of 0.01% to 5% by weight, wherein the composition has a particle size range of 0.1 microns to 60 microns. Furthermore, the invention relates to a process for preparing a seaweed composition comprising at least one seaweed and at least one agrochemically acceptable excipient in the form of an aqueous suspension. The invention also relates to a method of treating plants.	India (IN)
BR 11 2020 001535 0 [103]	Green algae, red algae, golden algae, brown algae, golden-brown algae, blue algae, blue-green algae.	Agricultural water-dispersible granular composition, process of preparing the same, its use and method of plant protection or improvement of health or yield	The invention relates to a granular seaweed composition comprising at least one seaweed and at least one agrochemically acceptable excipient selected from one or more surfactants, binders, or disintegrants having a weight ratio of seaweed to at least one surfactant, binder, or disintegrant in the range of 99:1 to 1:99. The seaweed comprises 0.1% to 90% by weight of the total composition. The composition has a particle size in the range of 0.1 microns to 60 microns. Furthermore, the invention relates to a process for preparing the granular seaweed composition comprising at least one seaweed and at least one agrochemically acceptable excipient. The invention further relates to a method of treating the plants, seeds, crops, plant propagation material, site, parts thereof, or the soil with the granular composition of seaweed.	India (IN)
BR 10 2018 072892 0 [129]	Seaweed red grapes of the phylum *Rhodophyta.*	Production process of edible, recyclable, disposable and biodegradable bio-cups and bio-utensils from agar-agar (hydrocolloid) extracted from Brazilian red marine algae of the phylum *Rhodophyta*	In the last two decades, disposable polystyrene cups made from petroleum have become widely consumed by companies due to their practicality, hygiene, and low price. This daily use represents an additional problem for the environment. The destination of this type of material is common garbage, as there is no recycling for disposable cups. The objective of the invention is to provide the food and biodegradable plastics industries with an innovative process for the production of edible, recyclable, disposable and biodegradable bio-cups and bio-utensils from agar-agar (hydrocolloid) extracted from Brazilian red grapes of the phylum Rhodophyta. It is intended for total or partial replacement of disposable and biodegradable utensils available on the market, such as cups, bowls, etc. Therefore, the type of technology described here is part of the development of new biodegradable materials, with a strong environmental appeal and with the objective of achieving environmental sustainability. When disposed of in the environment, bio-cups and bio-utensils suffer from natural degradation by microorganisms, as they are biodegradable products with superior characteristics to traditional products on the market, and which reduce the direct impact on the environment.	Brazil (BR)
BR 10 2018 075813 6 [122]	Red seaweed	Biodegradable organic bandages as an option in the treatment of burns or wounds and the respective process for obtaining	This application is aimed at the hospital sector, and refers to the development of a low-cost organic and biodegradable dressing capable of meeting healing needs, allowing wound monitoring and providing a viable product to health units, hospitals, and intradomiciles. The organic and biodegradable dressing is based on agar-agar, which is a hydrocolloid extracted from red algae, and an essential oil of Melaleuca that has antibacterial and antifungal activity. Thus, it is expected to provide the dressing with the necessary transparency for wound analysis, ideal consistency, as well as a more accessible cost.	Brazil (BR)
BR 11 2020 012394 3 [124]	*Rhodophyceae*, *Phaeophyceae*, and *Ascophyllum nodosum*	Identification and isolation method of bioactive compounds from seaweed extracts	It is a method of isolating and purifying bioactive compounds in an extract obtained from seaweed. The method involves the steps of: (a) circulating the extract through an ultrafiltration membrane that has an adequate molecular weight cut-off; (b) collecting filtrate from the extract to obtain a first filtrate fraction and a retentate; and (c) rinsing the retentate to obtain one or more additional filtrate fractions. The bioactivity of the first filtrate fraction and additional filtrate fractions can then be evaluated to determine their effectiveness on plant growth. One or more bioactive molecules isolated from an algal species are also described wherein one or more bioactive molecules have a molecular weight in the range of about 0.15 kDa to about 1 kDa.	France (FR)
BR 10 2019 003343 6 [104]	*Lithothamnium calcareum*	Compositions comprising seaweed and methods of using seaweed to increase animal product production	The present invention generally relates to a method for determining red algae inclusion rates in livestock feed and livestock supplements.	United States of America (USA)
BR 10 2019 003936 1 [125]	*Kappaphycus alvarezii*	Base mass/gel extraction process from seaweed while maintaining its properties.	The present invention consists of a manufacturing process for a base mass extracted in the form of a gel from the red macroalgae (*Kappaphycus alvarezii*) for application in food and cosmetics; the integral extraction of a polysaccharide with thiol groups has gelling and thickening properties at pH 4.5, promotes stable gel in a wide temperature range (0 to 60 °C), has an aqueous base with a slight odor and characteristic flavor, and is 100% natural. It has hydrating and healing properties (on burnt skin or under small cuts), in addition to humectant and refreshing characteristics, a matte effect, and a whitening effect, which can be used as a natural and safe raw material for the consumption as food, cosmetics, preparations for baths and the like, maintaining its stability at room temperature. It is a product free of preservatives, and maintainsits organoleptic characteristics for a period of up to one year in sealed packaging, without the addition of preservatives of any kind.	Brazil (BR)
BR 11 2020 021196 6 [128]	Not specified	Film forming system with a barrier effect, namely anti-atmospheric pollution, of natural origin and for use in cosmetics.	The subject of the application is a film-forming system, preferably for cosmetic use, consisting of at least one pregelatinized starch, and at least one starch-free polysaccharide chosen from gums of vegetable origin, preferably seaweed or seaweed gums plants, gums of microbial origin, and cellulosic derivatives. According to the invention, the film-forming system provides a barrier effect, namely anti-air pollution. The application also relates to a manufacturing process for such a film-forming system and to cosmetic compositions, namely topical ones, containing the said film-forming systems, such as skin care products, hair care products, make-up products, sun protection products, hygiene products, and fragrances.	France (FR)
BR 11 2020 021993 2 [132]	Not specified	Seaweed-elastomer composite, footwear component comprising the seaweed-elastomer composite and method for preparing a seaweed-based elastomer composite	An algae-elastomer composition that includes an elastomer matrix, algae, and a mixing additive sufficient to achieve a desired property. The algae may be present in a ground condition, having a particle size value between about 10 and 120 microns. The algae are mixed with the elastomer matrix in a dry condition having a moisture content of less than about 10%. A method for preparing the algae-based elastomer composite is provided, which includes the steps of pre-mixing an elastomer matrix; adding a load of seaweed; adding a mixing additive that includes a plasticizer; forming an elastomer-algae blend by blending the algae and elastomer at a temperature sufficient to be mixed further, wherein the temperature is about 10 °C higher than sufficient temperature for the elastomer alone; adding and mixing a curing or vulcanizing agent to the elastomer dispersing the elastomer-algae blend; and heating and curing the elastomer-algae blend into a final form.	United States of America (USA)
BR 10 2019 009021 9 [131]	Not specified	Simultaneous production of bio-hydrogen and bioethanol from algae carried off in discontinuous anaerobic reactors	The simultaneous production of biohydrogen and bioethanol from seaweeds in discontinuous anaerobic reactors. Faced with the awareness that the planet has finite resources and acknowledging the damage caused by pollution, alternative sources of energy that can compete with and/or replace the use of fossil fuels are sought. For the simultaneous production of hydrogen and ethanol by anaerobic fermentation, the residues of the macroalgae were used as a substrate, chemically pre-treated with an acid solution (H_2_SO_4_) at 1.0%, 1.5%, and 2.0% (*v*/*v*), with a pretreatment exposure time of 60 min, and without chemical pretreatment. The experimental procedure was carried out in reactors operated in batches maintained at mesophilic temperature (35 ± 1 °C) through the use of a rotating shaker incubator at 120 rpm, inoculated with UASB reactor sludge, and with the pH of the medium initially adjusted to 7.00. For higher acid concentrations, lower amounts of glucose and specific hydrogen production were quantified. The results for the crude reactors at 1.0%, 1.5%, and 2.0% H_2_SO_4_ (*v*/*v*) were 28.21, 97.42, 90.39, and 38.72 (mg/g of dried seaweed), and 5.80, 24.03, 1.99, and 2.53 mLH_2_/gSSVh, respectively.	Brazil (BR)
BR 11 2020 025226 3 [118]	*Saccharina latissimi*, *Laminaria saccharina*, *Laminaria digitate*, *Ascophyllum nodosum*, and *Laminaria hyperborean*	Antiparasitic composition, food product, and use of the antiparasitic composition	The present invention relates to an antiparasitic composition comprising at least one algal material, at least one plant material, or a combination of at least one algal material and at least one plant material for inhibiting, reducing, and/or suppressing the growth of at least one parasite.	Denmark (KB)
BR 11 2021 000776 8 [111]	Not specified	Method for improving water use efficiency and/or water productivity in plants and/or water management in agriculture.	The present invention relates to a composition based on seaweed and/or plant extracts for use in agriculture. It is objective improve water use efficiency, and the productivity in soil management, and water, resulting in an increased yield of cultivated plants per unit of water used.	Italy (IT)
BR 11 2021 002969 9 [109]	*Saccharina latissima* (*Laminaria* *saccharina*), *Laminaria digitate*, *Ascophyllum nodosum*, and *Laminaria hyperborean*.	Animal feed product and use of an animal feed product	The object of the present invention relates to an improved animal feed product comprising a seaweed material and/or a plant material with a reduced level of medicinal zinc.	Denmark (KB)
BR 10 2019 018113 3 [119]	*Prasiola crispa*	Pharmaceutical composition with fractions derived from seaweed and its use as a Leishmanicidal Agent	The present invention reveals the use of seaweed derivatives, with applications in the area of new leishmanicidal agents of natural origin. The invention describes the use of fractions of *Prasiola crispa* with a highly effective leishmanicidal effect, which provides a new non-toxic alternative for the treatment of this neglected disease, since it does not have cytotoxic effects on human cells.	Brazil (BR)
BR 11 2021 006877 5 [108]	*Phaeophytes*	Animal feed to enhance growth performance	Condensed soluble algae residues have been shown to be a beneficial feed ingredient for animal feed, in particular to enhance the growth of beef cattle.	European Patent Office (EPO); United States of America (USA)
BR 11 2021 007506 2 [120]	*Ulvales*	Use of a seaweed extract from the order Ulvales	The present invention refers to an extract of seaweeds of the order Ulvales that comprises sulfated and non-sulfated polyanionic polysaccharides, the molecular weight of which is less than or equal to 50 kDa, for use in the prevention and/or treatment of complications induced by post-traumatic immunosuppression.	France (FR)
BR 10 2019 026291 5 [110]	Not specified	Biodegradable glue for fixing small plants and uses	This technology deals with a non-toxic and biodegradable glue for fixing small plants, comprising corn starch (2 to 3% *w*/*v*); maltodextrin (2 to 3% *w*/*v*); gums (4 to 25% *w*/*v*); and 0.2 to 0.32% *w/v* montmorillonite clay. The glue may comprise one or more preservatives, such as potassium sorbate (0.01 to 0.05% *w*/*v*), sodium chloride (0.01 to 0.05% *w*/*v*), or abscisic acid (5 to 10 µM). Its composition may also contain the inoculum of small plants, such as propagules of bryophytes, cyanobacteria, algae, and mycorrhizal fungi; in addition to nutrients. The glue allows the plants to adhere to the substrate, retains moisture during transplantation, and can be used in landscaping or in the reintroduction of small plants in a natural habitat by fixing them on rocks, trunks, and other substrates.	Brazil (BR)
BR 10 2019 027698 3 [123]	*Hypnea e Dictyopteris*	Dilutor for cryoprotection and freezing of gametes based on seaweed	The present invention deals with an algal plant extract as a cryoprotective extender as a substitute for an animal origin extender, which can be used in animal reproduction programs of utility in research and commercial interest. The present invention is in the field of Biotechnology and natural products for veterinary or human use. Its use may be preferred in the process of cryopreservation of both fresh and frozen gametes.	Brazil (BR)
BR 11 2022 004774 6 [116]	Seaweed selected from the families of *Gigartinaceae*, *Bangiophyceae*, *Palmariaceae*, *Hypneaceae*, *Cystocloniaceae*, *Solieriaceae*, *Phyllophoraceae*, *Furcellariaceae,* or combinations thereof.	Seaweed-based composition and food product, beverage, nutritional product, dietary supplement, feed product, personal care product, pharmaceutical product, or industrial product	The present invention relates to a seaweed-based composition comprising water, a seaweed powder and an additional component, with the said additional component being chosen from the group consisting of glucomannans, galactomannans, native starch, and combinations thereof. The red seaweed is preferably selected from the families of *Gigartinaceae*, *Bangiophyceae*, *Palmariaceae*, *Hypneaceae*, *Cystocloniaceae*, *Solieriaceae*, *Phyllophoraceae*, *Furcellariaceae*, or combinations thereof.	United States of America (USA)
BR 11 2022 014527 6 [134]	*Asparagopsis taxiformis* and *Asparagopsis armata*	Bioreactor and seaweed culture method	The present invention relates to a bioreactor, which includes a first compartment designed to retain seaweed sporophytes; a second compartment in fluid communication with the first compartment. Also, includes one or more seating surfaces, and a first porous barrier between the first and second compartments. The second compartments allow seaweed spores to pass from the first compartment to the second compartment, and to systems comprising the bioreactor. Also provided herein are methods of culturing seaweed, for example using a bioreactor.	United States of America (USA)
BR 11 2022 020495 7 [107]	*Asparagopsis taxiformis*	Compositions comprising seaweed and methods of using seaweed to increase animal product production	The present invention generally relates to a method for determining red algae inclusion rates in livestock feed and livestock supplements.	United States of America (USA)
BR 11 2022 025357 5 [135]	Not specified	Systems and methods for growing algae using direct air capture	It is a system together with a method of supplying an algae-growing fluid with nutrients (e.g., carbon dioxide and nitrogen) directly from the atmosphere. Supplying nutrients directly from the atmosphere reduces operating costs and environmental impacts, as well as providing greater flexibility in the location of algae farms.	United States of America (USA)
BR 11 2022 026526 3 [105]	*Ulva armoricana, Solieria cordalis*, and *Euchema spinosum*	Formulation and use of a composition for plant health	The invention relates to the use of an organomineral composition by foliar application to stimulate plant development in the presence of at least one abiotic stress, with the said composition comprising the following compounds: 5 to 50% of seaweed extract; 0.5 to 2.5% of soluble silica expressed in SiO_2_; 0 to 30% mineral nutrient salts; 0 to 12% of trace elements; and 0 to 30% of organic acids. The percentages are expressed as the weight of the dry matter of each of the said compounds in relation to the total weight of the dry matter of the organomineral composition.	France (FR)
BR 11 2022 025481 4 [127]	Red seaweed (*Rhodophyceae*), belonging to the families *Gigartinaceae*, *Hypneaceae*, *Furcellariaceae*, and *Polyideaceae*.	Composition, in particular cosmetic composition, and cosmetic process for caring for keratin materials, especially the skin.	The present invention relates a cosmetic composition for skin containing keratin, comprising between 0.5% and 4% weight of short-chain fatty acid salt and, at least, 90% by weight of an aqueous phase relative to the total weight of the composition. It is gelled by a hydrophilic gelling agent chosen from (i) at least one polymer chosen from polyacrylamides and 2-acrylamido-2-acid polymers and copolymers methylpropanesulfonic acid, (ii) at least one polysaccharide produced by microorganisms or isolated from algae, (iii) at least a cellulose or one of its derivatives, (iv) at least one carboxyvinyl polymer, or (v) one of its mixtures. It also refers to a cosmetic process for caring for keratin materials, the skin in particular.	France (FR)
BR 10 2021 014827 6 [121]	Not specified	Production process of polymeric nanoparticles of R-Phycoerythrin and resulting products with therapeutic and diagnostic potential against cancer	*R*-Phycoerythrin (R-FE) is a protein pigment that is produced by red algae, which has spectroscopic properties of absorption and the emission of light at three specific wavelengths, which is why the pigment can be photosensitized inside the cell by photodynamic therapy, producing reactive molecules that have a positive effect on suppressing cancer cells. To increase the permeability of this molecule in cells, as well as its accumulation in the cell interior, a production process of R-FE polymeric nanoparticles coated with polyvinyl acetate and polycaprolactone was developed through the double emulsification by the evaporation of the organic solvent.	Brazil (BR)
BR 10 2022 006876 3 [106]	Brown seaweed	Phytoregulator for plants	A phytoregulatory composition including L-proline, kaolin, and, optionally, brown algae, which is provided in wettable powder form for use on plants in the flowering and fruiting stages.	United States of America (USA)

### 2.2. Seaweeds from the Brazilian Coast: Environmental Aspects

Prospecting chemical species in this Brazilian marine ecosystem is relevant both from a biological and an environmental point of view. However, despite their potential, marine organisms suffer from the ecological pressures derived from anthropogenic actions, which might lead to the bioaccumulation of potentially toxic elements and microplastics, as well as changes in the trophic structure, impacting promising species with bioprospecting potential. Furthermore, climate change is another factor to be considered in the monitoring of chemical species. The trend of accumulation of microplastics, potentially toxic elements, and other emerging contaminants (drugs and pesticides, among others) in seaweed is worrying due to its increasing consumption, mainly in Asia, Europe, and the Americas. The accumulation capacity depends on several factors, such as the type of algae (brown, red, or green), exposure to waves, location, temperature, salinity, light, pH, the season of the year, metabolic processes, and the affinity of the plant medium for each chemical species [137,138]. 

Searches were carried out for articles published that correlated microplastics (MP) with seaweed in Brazil, within the 20-year interval, with an emphasis on the period of 2017–2023. No studies conducted in Brazil with this theme were found. Therefore, there is still a gap in the literature with regard to the discussion of this relevant topic, since little is known about the impact of the presence of MP on seaweeds. *Cystoseira barbata* alginate has a strong adsorptive capacity for potentially hazardous metals, also known as heavy metals. Studies using lead and copper were conducted using this seaweed, and the results demonstrated good adsorption capacity, intermediated by the alginate [139]. Seaweeds act as excellent adsorbents for chemical substances such as dyes. Some characteristics that support this effect include the presence of pores on the seaweed’s surface, the surface area of these pores, and the presence of functional groups. Due to the electrostatic and hydrogen bonding between the dye and the seaweed surface, these groups coordinate the adsorption process, resulting in chemisorption [140]. 

Sundbæk et al. [141] carried out studies with *Fucus vesiculosus,* and demonstrated that, in the presence of MP, this species has the capacity to adsorb this contaminant. They concluded that, in addition to adsorbing MP on its surface, this species has pheoficial hair and cell walls rich in alginate, which contributes to the adherence of this material. According to Li et al. [142], some seaweeds have the ability to adsorb MP, which makes them vectors of contamination with this pollutant. This is worrying because many species are used for the manufacture of Nori, which is consumed by humans.

Based on this discussion, it was agreed that further studies involving the impact of MP associated with seaweed and corals, in Brazil and worldwide, are necessary. On the other hand, even with this gap in the literature, some studies were found that highlighted the presence of MP on Brazilian beaches and in native marine species that were published in the last five years. This leads us to reflect on the potential presence of these materials in marine algae and corals. 

Studies have shown MP contamination on several beaches in different regions of Brazil [143,144,145,146,147,148,149,150,151]. These MP are commonly found in sediments [152] and some marine organisms, such as anemones, fish, oysters, and rays [143,153,154,155]. The main polymers that constitute the MP that were identified on Brazilian beaches and marine organisms were polyethylene, polypropylene, polyamide, polystyrene, polyurethane, polyester, polyethylene terephthalate, styrene, and high-density polyethylene [143,145,146,147,149,150,153,155,156]. In addition to the MP, potentially toxic elements are considered inorganic pollutants of great relevance to public health and the environment. These elements tend to accumulate in water and in marine organisms (including seaweed), until they reach toxic concentrations, causing damage to tissues and other structures (including humans), expressing particular symptoms that indicate the changes that have occurred in that environment [153,157]. The accumulation capacity depends on several factors, such as the type of algae (brown, red, or green), exposure to waves, location, temperature, salinity, light, pH, season, metabolic processes, and the affinity of the plant medium for each element [137,138]. 

Despite generating products that are beneficial to human health, seaweeds are an important source of exposure to potentially toxic elements [158]. Contamination of the ecosystem with elements such as arsenic (As), cadmium (Cd), mercury (Hg), lead (Pb), and others (including those considered essential, in high concentrations, such as Cu, Mn, Zn, etc.) is considered very dangerous, as these do not play any biological or nutritional role in animals, plants, or microorganisms. Some elements are naturally formed in the Earth’s crust, but the inadequate treatment of industrial and domestic effluents, the leaching of pesticides and fertilizers in agriculture, maritime transport, the exploitation of mineral resources on the continental shelf, and the direct discharge of contaminants carried out by submarine outfalls have caused the concentration of these elements to increase significantly in recent decades, especially in the oceans [159].

Marine organisms, including seaweed, have been widely used to indicate and/or monitor changes in the environment where they live and the conditions to which they are continuously exposed [160]. The concentration of elements, whether essential (macro- and microelements) or potentially toxic to these organisms is significantly higher when compared to water, and accumulates over time. Therefore, it is of great interest to understand the mechanisms and quantify these elements that increase substantially along the food chain and, therefore, that can cause adverse effects on human health. 

Mercury, when in contact with seawater and living organisms, is rapidly transformed into methylmercury (CH_3_Hg^+^). This compound, easily absorbed by biological membranes, accumulates in the nervous system, causing negative neurological and reproductive effects in animals [161]. Cd is strongly retained in the living organism, inhibits the action of antioxidants, and disrupts Fe and Zn-dependent metabolic pathways, decreasing growth rates and the reproductive system. When absorbed by humans, it accumulates in the kidneys, causing irreversible damage to the renal system [157]. As can be found in two oxidation states, As^3+^ (arsenite) and As^5+^ (arsenate), in addition to other organic structures. The toxic effect on the living organism depends on these states, with arsenite being about 60 times more toxic than arsenate. In high concentrations, As inhibits enzymatic action and blocks cellular respiration, which increases the risk of cancer in humans [162]. Pb interacts with metallic ions and sulfated groups that are involved in several enzymatic processes, inducing the production of reactive oxygen species. In humans, this element mainly affects children and pregnant women, hindering the development of the fetus, in addition to causing problems in the gastrointestinal tract and Alzheimer’s disease [157,163]. Antimony is considered a toxic chemical element of global concern that has unknown biological functions. Its presence in aquatic environments is mainly due to anthropogenic activities, but is also a result of natural processes, such as rock weathering and soil runoff. The level of SbV depends on the redox status of the aquatic environment, and SbIII is characteristic of water with a low oxygen content [84]. Manganese (Mn) in high amounts causes toxic effects on the central nervous system. Copper (Cu) and Zn can become toxic when present in high concentrations in the environment, producing free radicals in tissues and causing irreversible damage to some species. For humans, excess Cu intake can cause Menkes disease, a disorder of genetic origin that leads to rapid brain degeneration, usually accompanied by seizures, hypothermia, and growth retardation [78]. 

### 2.3. Corals from the Brazilian Coast: Bioactive Compounds

The coast of Brazil has the only truly coral reefs in the South Atlantic, which are characterized by low scleractinian coral species diversity but high endemism [164,165]. These reef communities are characterized by occurring in high turbidity waters, with distribution from the Parcel de Manuel Luis (about 1° S) to Abrolhos (about 18° S), and occasional occurrences at the southwest and south coasts of Brazil. The endemic species are distinguished into two categories: (1) species that were related to Caribbean coral species but lost connection when the uplift of the Andes mountains changed the course of the Amazon river, creating a plume that acts as a barrier to prevent larval dispersion due to its lower salinity, leading to isolated coral communities [166]; and (2) ancient species of the genera Mussismilia that survived in the South Atlantic during the Pleistocene sea level lowering due to refuges created by the Abrolhos seamounts and the Vitória-Trindade mountains, and that only exist in Brazil [164].

These two phenomena are responsible for the high endemism of the Brazilian coral fauna and, consequently, its worldwide importance, even with low diversity compared to other areas such as the Caribbean and the archipelagos of Southeast Asia. Such a unique characteristic reveals the potential of these corals for bioprospecting and the importance of conservation. So far, 23 species of scleractinian corals have been described off the coast of Brazil, including two invasive species, *Tubastraea tagusensis* and *T. coccinea*. Another group of stony corals are the hydrocorals, which have five endemic species in the Brazilian marine waters [167]. 

An important group of corals, particularly for chemical compound investigation, are the black corals and octocorals. So far, four species of black corals were recorded off the coast of Brazil [168], and are grouped in two genera: Antipathes (three species) and Cirripathes (one species). However, black coral diversity has not been properly investigated in Brazil [164]. The octocorals group has about 100 species or morphotypes that have been reported off the coast of Brazil, from which 14 species are endemic to the Brazilian coast [169]. Octocorals are divided into deep-water corals and shallow-water corals, of which the diversity of the latter is well-known when compared to the later.

The exploration of the potential of secundary metabolites produced by stony corals from the coast of Brazil with applications in different areas (the pharmaceutical, chemical, and cosmetic industries) still requires a thorough investigation. A few studies have addressed or successfully achieved the identification of potential compounds in corals. Recently, Romanelli et al. (2022) isolated and characterized an indole alkaloid (Figure 10) named 6-bromo-2′-de-*N*-methylaplysinopsin (**42**) (Figure 10) from the invasor coral *Tubastraea tagusensis*, which has proved to be effective against *Trypanosoma cruzi* [170]. Another work that explores the potential of corals from Brazil was carried out by Lhullier et al. in 2020, which involved the testing of the antiproliferative effects, and the antiprotozoal, anti-herpes, and antimicrobial activities of the organic extracts of 14 corals from northeast Brazil [171].

Octocorals, on the other hand, have proved to be an important source of novel secondary metabolites, providing an extensive diversity of compounds such as steroids, acetogenins, sesquiterpenes and diterpenes [172], which are mostly derived from the mevalonate pathway [173]. The origin of such a high diversity of compounds in octocorals may be associated with ecological interactions. In addition, to the fact that octocorals do not have a physical defense such as stony corals, such as the cnidocytes cells, and these compounds may act as chemical defenses [174]. A very detailed review of secondary metabolites extracted from octocorals from Brazil were made [172], where 106 compounds were isolated from 20 species and primarily consisted of steroids, diterpenes, sesquiterpenes, and prostaglandins. The main biological activity from these compounds reported by this review were associated with their antimicrobial, antiproliferative, antipredator, antifouling, cytotoxic, and antiprotozoal properties.

A considerable advance in compound diversity in octocorals from Brazil and their pharmaceutical application has already been developed, but regarding the stony corals (scleractinia), and considering its biodiversity characteristics (i.e., high endemism), a few studies have properly addressed the potential of this group as a source of promising chemical compounds. It is important to promote the investigation of secondary metabolites on these marine resources, which can offer potential applications in different fields, leading to promising molecules that can be used for the treatment of pathogenic agents that are still harmful to the human population, especially in the tropics, as previously explored [170].

### 2.4. Corals from the Brazilian Coast: Environmental Aspects 

Contaminants in corals can be accessed and evaluated in different matrices, such as the coral tissue (i.e., the coral polyp or the organic matrix) and the coral exoskeleton (i.e., the aragonite mineral or the inorganic matrix), in which the coral tissue can record the present levels of contaminants in the short-term, whereas the coral skeleton can record the levels of contaminants at long-term scales (decades or centuries).

Regarding the contamination accessed by the coral tissue, inorganic contaminants such as the potentially toxic elements (PTEs) that exist as dissolved or particulate forms [175] can be incorporated in the coral tissue by ingestion (heterotrophy) or directly from the seawater column [176]. The rate of accumulation of PTEs in corals is governed by the combination of coral physiology and environmental availability [177], which can change over time. For example, in 2015 Metian et al. [176] accessed the seasonal availability for the PTEs As, Cd, Cr, Pb, and Zn in corals from the species *Astrangia poculata* in the northeastern United States, and indicated a fluctuation in the levels of these PTEs throughout the seasons of the year, showing that the coral tissue can report the variability of contaminants according to the local seawater conditions.

However, in the coral tissue matrix, investigations on organic chemical contaminants such as petroleum hydrocarbons (especially polycyclic aromatic hydrocarbons—PAHs) [178] organic ultraviolet (UV) filters [179], and persistent organic pollutants (POPs) [180] have highlighted the urgent need to evaluate the levels of these contaminants on corals worldwide.

PAHs are one of the organic chemical class compounds that have raised the most concern about their impact on coral reef environments, especially because of their persistence in the environment and their bioaccumulation and bioconcentration properties, together with their toxicity and harmful potential to act as carcinogenic and mutagenic agents [181]. In recent decades, the impact of oil spillage on marine ecosystems, including coral reefs, has drawn attention to the impact of petroleum-derived polycyclic aromatic hydrocarbons. For example, from August 2019 to January 2020, coastal marine ecosystems in the northeast and southeast of Brazil have been impacted by large amounts of crude oil that extended through an area of about 3000 km [182], and the impact of this disaster is still under investigation.

POPs are another important class of emerging contaminants that impact corals, and they are also present in pharmaceutical products such as ultraviolet (UV) filters, which may include various compounds such as benzophenone, cinnamates, and chrylene derivatives [183]. These filters can be used for personal care, but can also be used in paints and in the plastic industry to reduce the photodegradation of products [184]. Some of these contaminants were associated with harmful effects such as coral bleaching and planula larvae deformities [185,186]. 

In the last five years, no work has directly addressed the evaluation or monitoring of contaminants in the coral tissue matrix. However, the assessment of PAHs in the coral tissue and skeleton is underway, which will soon reveal the status of PAHs contamination on corals from the Todos os Santos Bay in the state of Bahia. However, many aspects regarding the influence of contaminants (e.g., emergent contaminants and PTMs) on the extensive coastal territory that corals inhabit in Brazil still need to be addressed. Most of these territories are under high pressure due to human activities such as tourism, environmental degradation within metropolitan areas, and agriculture expansion, among others.

Contaminants can also be evaluated at the coral exoskeleton. They has an ad-vantage over the organic matrix, as the coral exoskeleton can record pollutants during their life period. Thus, generating ordinated and sequential records that can be assessed to re-construct past environmental conditions at a decadal or even centennial scale, revealing the environmental history of the tropical ocean history [187,188,189]. Among the pollutants recorded in the corals’ exoskeleton are potentially toxic elements (PTEs), as well as organic compounds such as PAHs [181].

Metal concentration and stable isotope variations in the coral exoskeleton can be used to reconstruct the past environmental conditions of the surrounding waters, which can be accessed at seasonal, annual, and decadal scales [190]. Such reconstructions are mostly associated with seasonal variability in riverine discharge related to heavy precipitations that increase sediment input on reef systems, or episodic events such as dam failures or civil construction activities that discharge highly metal-concentrated sediment into marine environments [191]. 

Metallic ions can be incorporated into the coral exoskeleton mostly by the following mechanisms: (i) replacing Ca^2+^ as trace elements in the aragonite crystal lattice due to its ionic radius compatibility and charge balance; (ii) by adsorption to the skeletal surface, (iii) by the retention of particulate material in the cavities of the coral skeleton; and (iv) by the association with organic matter [191]. These characteristics enable the use of geochemical records incorporated in the coral exoskeleton as environmental and climate indicators.

In Brazil, studies carried out by Cardoso et al. [192] in 2022, and Evangelista et al. [193] in 2023, have evaluated the concentrations of metals in the coral exoskeleton of the Abrolhos Bank. These studies attempted to assess the impact of the Fundão dam collapse in 2015, which carried over about 50–60 million cubic meters of ore tailings with several PTEs to the Doce River, which discharged a sediment plume into the South Atlantic Ocean and transported it via ocean currents to the most important reef system in the South Atlantic, the Abrolhos coral reef [167]. 

Another approach regarding the use of coral exoskeletons for the reconstruction of past environmental conditions is the application of Isotope Ratio Mass Spectroscopy (IRMS) to coral carbonate to evaluate the carbon and oxygen isotopic composition recorded in the coral skeleton [188]. The oxygen stable isotope (δ^18^O) signatures in corals have been widely used to reconstruct the SST history of the tropical ocean [194,195], improving our understanding of global warming and climate change in the tropical oceans.

The carbon stable isotopes (δ^13^C) of corals have been used to evaluate changes in the ocean surface DIC [196,197,198], which is affected by changes in the δ^13^C of the atmospheric CO_2_ provoked by the input of anthropogenic CO_2_ to the atmosphere from fossil fuel burning and deforestation. This long-term anthropogenic change in the carbon isotopic composition of atmospheric CO_2_ is known as the Suess Effect [199]. For the last five years, in Brazil, only the study in 2018 by Pereira et al. [200] has reported a systematic change in the δ^13^C of corals from the Rocas atoll, a pristine oceanic island that is associated with the global increase of CO_2_ resulting from anthropogenic origin.

## 3. Conclusions and Perspectives

Herein are reported the secondary metabolites and biological activities of seventy-one seaweed species from the Chlorophyta, Rhodophyta, and Ochrophyta phyla, in addition to fifteen corals. Despite the range of investigated biological activities, most of the studies focused on the screening of various extracts, but did not fully chemically characterize them, or relied on spectroscopic methods to suggest the presence of compound classes such as carotenoids and polyphenols. Only a few studies targeted the isolation of compounds, which included one alkaloid from the coral *Tubastraea tagusensis*. In this sense, the number of newly reported compounds is fewer than expected in light of the diversity represented by the seaweeds and corals in Brazil. 

There are currently about 750 seaweed species registered in Brazil, but the number of species and occurrence reports are constantly growing [201]. In addition to the diversity status, there are two main zones proposed for the Brazilian seaweeds based on the distribution of the species: a tropical zone in the northeast region and a warm temperate zone ranging from Rio de Janeiro state to the extreme south of the country. In between, a transition zone with particular environmental characteristics exists in Espirito Santo State [202]. These combined aspects provide a promising perspective related to the hidden chemical diversity of Brazilian seaweeds, which can be influenced by the diversity of species, ecological interactions, and the different abiotic stressors faced by the organisms across the Brazilian coast. More accurate chemical information about the secondary metabolites of Brazilian seaweeds is needed, and might represent a challenge to the field of seaweed-derived natural products in Brazil. Although the low number of isolated compounds, screening, and prospection studies are crucial to opening new paths and represent a powerful tool for the conservation of species, they often describe new organisms and their chemotaxonomic features [203]. Additionally, these studies can guide the prioritization of target organisms with aggregated biotechnological potential. A successful example of a prospection program was developed by the United States National Cancer Institute (NCI-USA), and yielded the discovery of anticancer drugs such as taxanes and camptothecins [204]. Thus, novel workflows are required to spur the discovery of novel metabolites produced by Brazilian seaweeds. Notably, a growing number of studies are using hyphenated techniques such as gas or liquid chromatography coupled with mass spectrometry to chemically profile Brazilian seaweeds. These metabolomic approaches offer advantages such as the high accuracy of data and the possibility of exploring a large number of organisms and extracts concurrently [205], and they have applicability to target compounds exercising ecological roles, chemotaxonomically relevant compounds, compounds produced in response to changing abiotic factors, and bioactive compounds. In the face of the Ocean Decade announced by the UN in 2021, it is necessary to increase collaborations across different fields through a multidisciplinary approach to advance our knowledge of the potential of Brazilian seaweeds and their role in the pursuit of sustainable development.

Seaweeds from Brazil are a potential source of microelements and other compounds that have beneficial effects [206]. However, new analytical approaches are needed to describe their amounts. Besides microelements, there is a literature gap regarding the presence of potentially toxic elements and other emergent contaminants in seaweeds and corals from the Brazilian coast. These contaminants are targets for the evaluation of food safety and anthropogenic impact since they offer risks to human and animal health, in addition to the marine environment. 

## Figures and Tables

**Figure 1 molecules-28-04285-f001:**
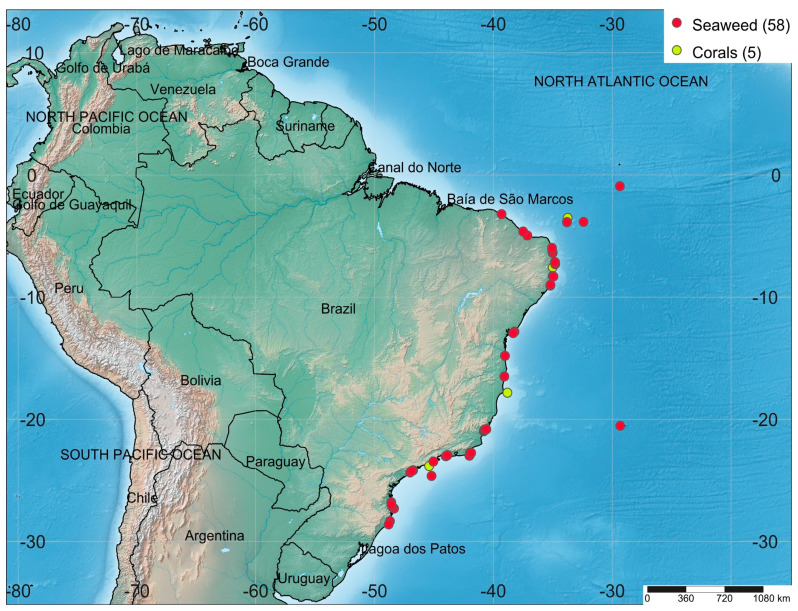
Collection spots of the seaweeds and corals collected along the Brazilian coast and reported in the literature in the period ranging from 2018 to 2022. Numbers represent the total of collection sites. Map created by SimpleMappr software (https://www.simplemappr.net, accessed on 25 April 2023) using the GPS coordinators (yellow—corals; red—seaweed collection sites) cited in the analyzed manuscripts.

**Figure 2 molecules-28-04285-f002:**
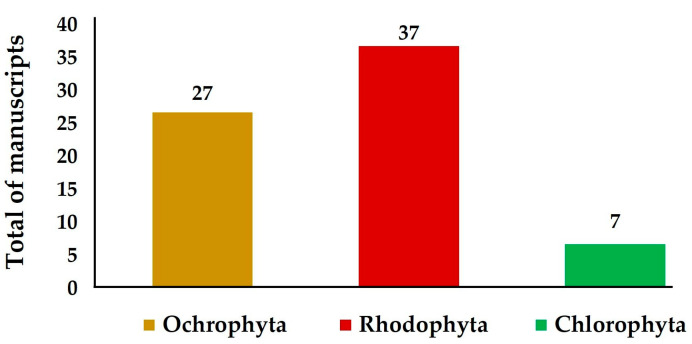
Seaweed species in each phylum investigated in the reports, addressing their chemical composition and bioactivities in the period ranging from 2018 to 2022.

**Figure 3 molecules-28-04285-f003:**
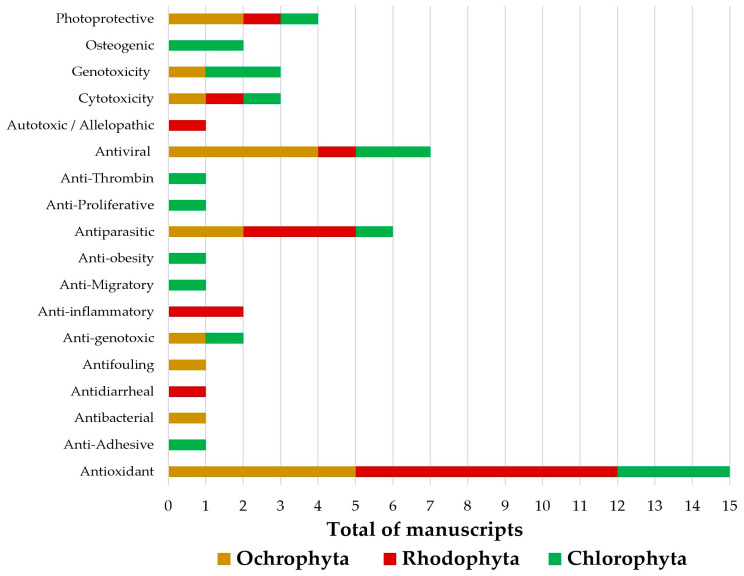
Biological activities investigated in the reports addressing the biotechnological potential of seaweeds from the Brazilian Coast in the period ranging from 2018 to 2022.

**Figure 4 molecules-28-04285-f004:**
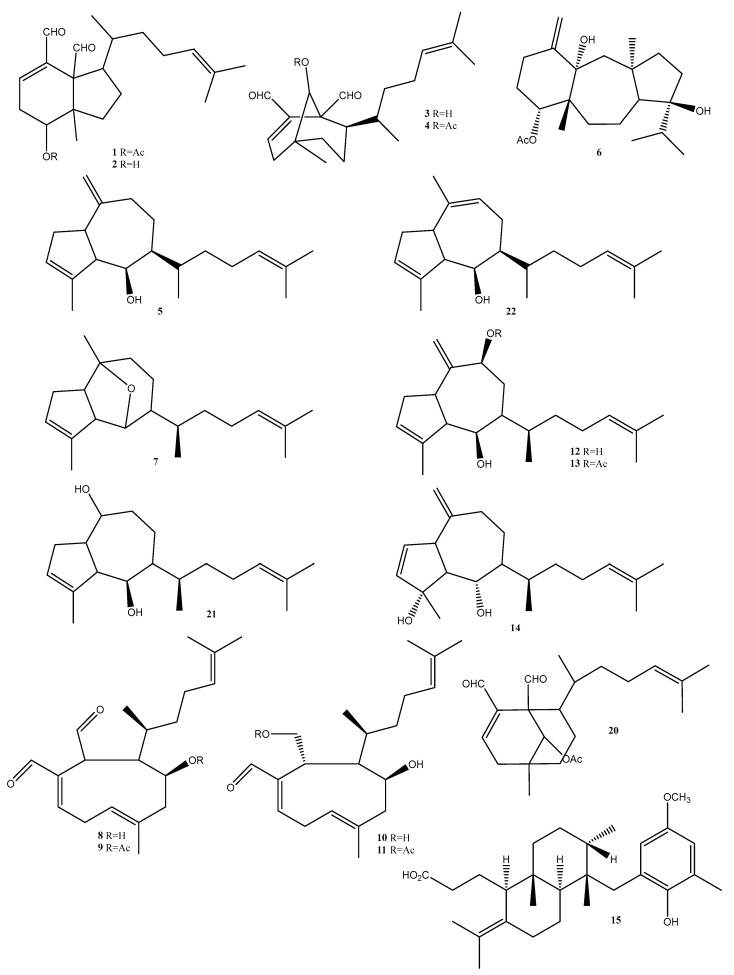
Compounds reported from seaweeds of the *Dictyota*, *Canistrocarpus*, and *Stypopodium* genera collected on the Brazilian coast in the period ranging from 2018 to 2022. Numbers represent the order of compounds as shown in the text. Compounds **1**–**4** (diterpenes from *D. menstrualis)*, **5** (pachydictyol A), **6** (dolastane), **7** (dictyoxide), **8** (4b-hydroxydictyodial), **9** (4b-acetoxydictyodial A), **10** (18,4-dihydroxydictyo-19-al A), **11** (18-acetoxy-4-hydroxydictyo-19-al), **12** (dictyol B), **13** (dictyol B acetate), **14** (dictyotadiol), **15** (atomaric acid), **20** (9-acetoxydichotoma-2,13-diene-16,17-dial), **21** (dictyol C), **22** (isopachydictyol A).

**Figure 5 molecules-28-04285-f005:**
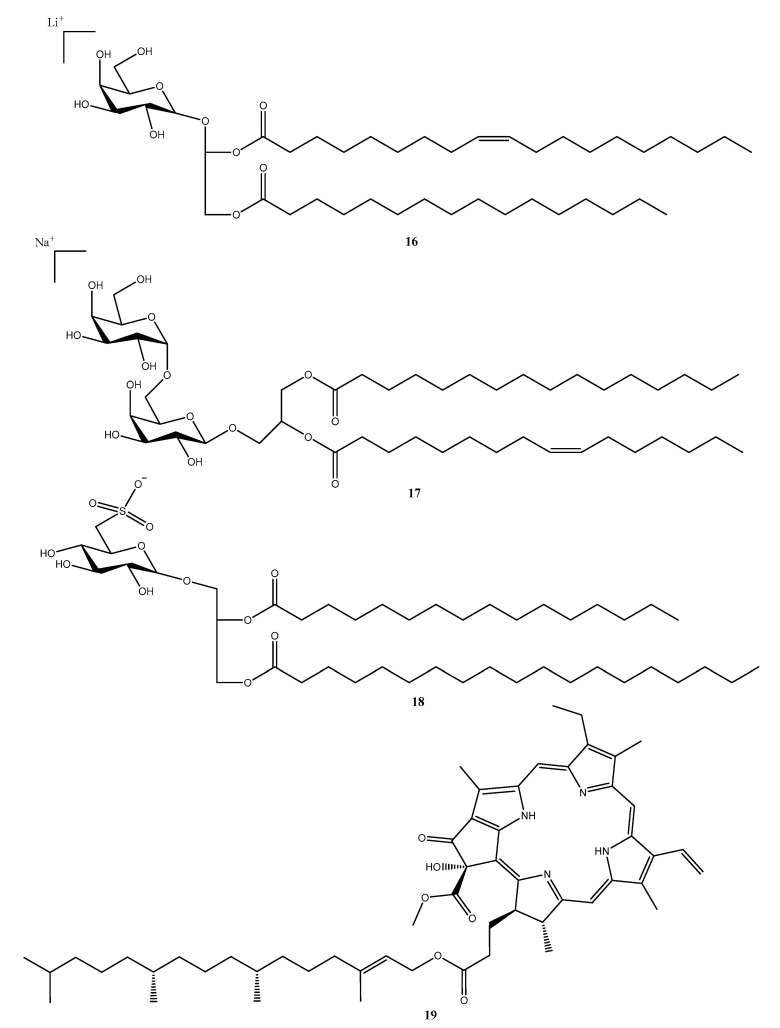
Compounds reported from seaweeds of the *Sargassum* genus collected on the Brazilian coast in the period ranging from 2018 to 2022. Numbers represent the order of compounds as shown in the text; Compound **16** (monogalactosyldiacylglycerols), **17** (digalactosyldiacylglycerols), **18** (sulfoquinovosyldiacylglycerols), and **19** (Pheophytin Sp-1).

**Figure 6 molecules-28-04285-f006:**
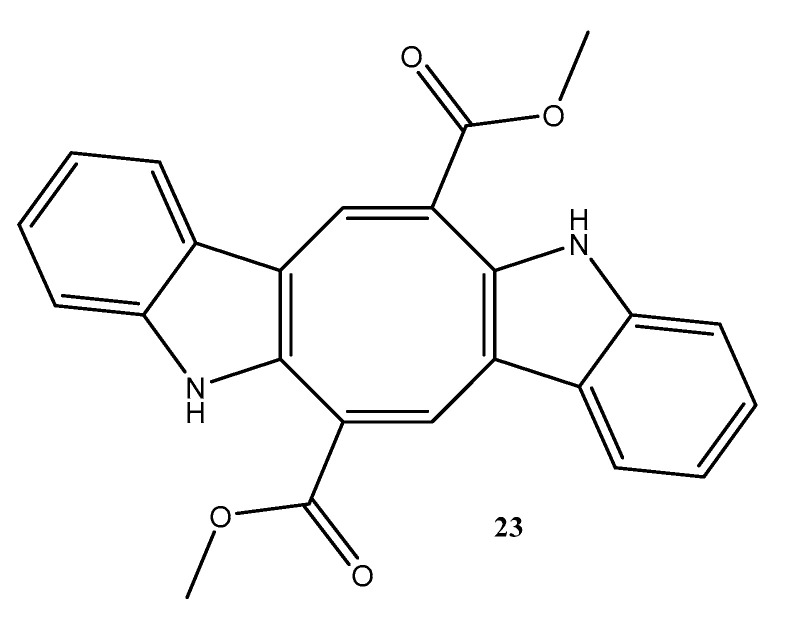
Compounds isolated from the seaweed *Caulerpa racemosa* collected on the Brazilian coast and reported in the period ranging from 2018 to 2022. The number represents the order of compounds as shown in the text. Compound **23**, caulerpin.

**Figure 7 molecules-28-04285-f007:**
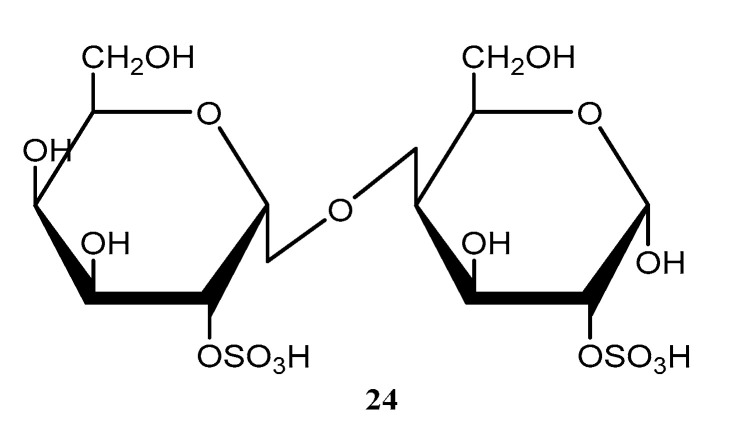
Compound isolated from the seaweed *Digenea simplex* collected on the Brazilian Coast and reported in the period ranging from 2018 to 2022. The number represents the order of compounds as shown in the text. Compound **24**, a sulfated polysaccharide.

**Figure 8 molecules-28-04285-f008:**
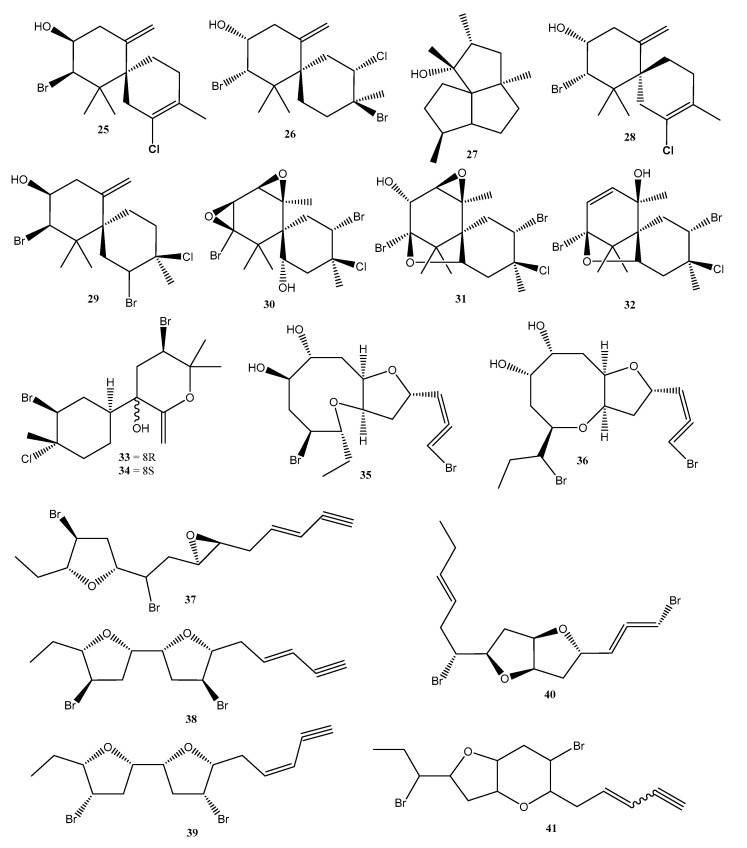
Compounds reported in seaweeds of the *Laurencia* and *Plocamium* genus collected on the Brazilian coast and published in the period ranging from 2018 to 2022. Numbers represent the order of compounds as shown in the text. Compounds **25** (5-chloro-1-(*E*)-chlorovinyl-2,4-dibromo-1,5-dimethylcyclohexane), **26** (elatol), **27** (obtusol), **28** (silphiperfolan-7β-ol), **29** ((−)-rogiolol), **30** (prepaciphenol epoxide), **31** (johnstonol), **32** (pacifenol), **33** (laucapyranoid B), **34** (laucapyranoid C), **35** (laurendecumallene A), **36** (laurendecumallene B), **37** (laureepoxide), **38** ((3*E*)-elatenyne), **39** (elatenyne), **40** (kumausallene), and **41** (laurobtusin).

**Figure 9 molecules-28-04285-f009:**
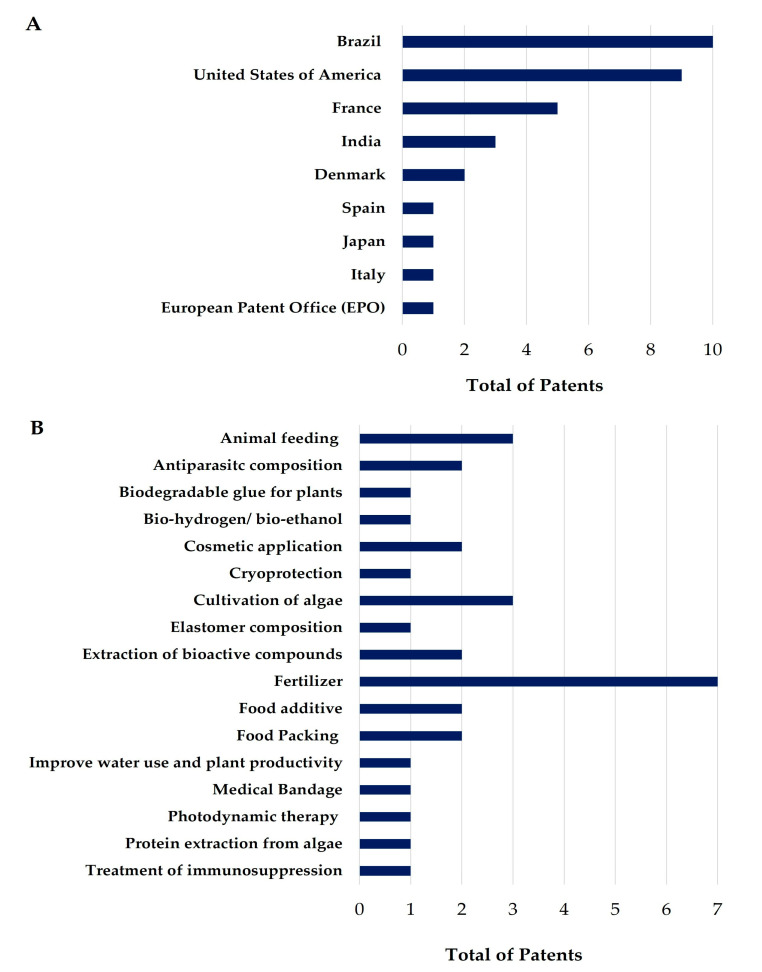
(**A**) Number of patents deposited by countries of origin of the depositors; (**B**) Number of patents by the registered use/method of application.

**Figure 10 molecules-28-04285-f010:**
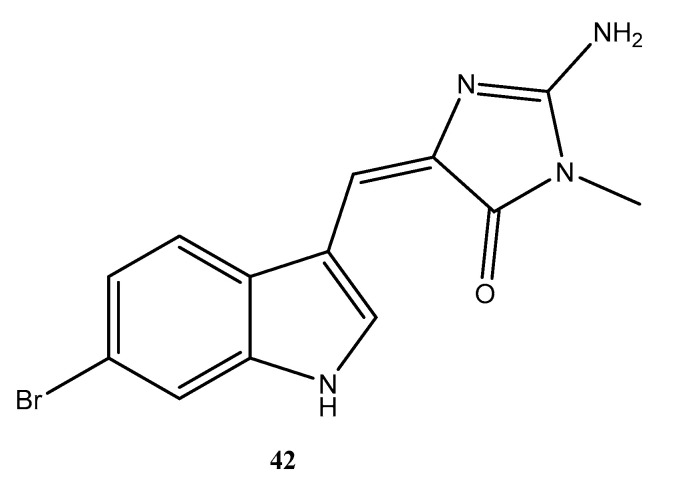
Compound (**42**) the 6-bromo-2′-de-*N*-methylaplysinopsin isolated from the coral *Tubastraea tagusensis*.

## Data Availability

No new data was created or analyzed in this study.

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
