# Peer review of "Seaweeds and Corals from the Brazilian Coast: Review on Biotechnological Potential and Environmental Aspects"

_molecules, 2023, doi:10.3390/molecules28114285_

Round 1
Reviewer 1 Report
1. The title should be:
Brazilian Seaweeds and corals: Review on Biotechnological potential and environmental aspects;
2. The abstract is too long, shorten it and a maximum of 200 words;
3. SciFinder is a Compounds by ACS database, and it is not suitable as a detailed Species finder. fix it.
4. The placement of all the title figures is wrong, it should be under the figure. and provide details of each image obtained from where? what software do you use? what about permissions and licenses to use it?
5. Figures 4, 6, 7, 8 are not clear, what compounds are they? why is there no indication of the names of each molecule? and where did this image come from? what software do you use? what about permissions and licenses to use it?
6. Almost everything discussed is seaweeds or algae, but in the title there is "corals" and a few appear and add "corals" comprehensively.
Those are all major and basic problems, if they are not corrected properly, then they will not meet the scientifically soundness criteria!
Please check grammar and spelling
Author Response
Dear Reviewer #1:
Thank you for allowing us to revise our manuscript entitled “Seaweeds and corals from the Brazilian coast: Review on biotechnological potential and environmental aspects” (ID: molecules-2398683). We thank the reviewers for the careful lecture of our manuscript and thoughtful comments, which certainly have improved it. We would like to say that we accepted all comments raised by the reviewers. We have addressed each reviewer comments, as described below. We would like to thank you for your kindly to let us revise our previous version. We hope that the manuscript is now suitable for publication.
A review of the language, spelling and grammar of the manuscript was carried out.
All changes are highlighted in red color, in the submitted text!
Best regards,
- The title should be: Brazilian Seaweeds and corals: Review on Biotechnological potential and environmental aspects;
Response: Thanks to the reviewer for the suggestion! The title has been changed as suggested!
- The abstract is too long, shorten it and a maximum of 200 words;
Response: Thanks to the reviewer for the suggestion! The abstract was adequate for 200 words as suggested!
- SciFinder is a Compounds by ACS database, and it is not suitable as a detailed Species finder. fix it.
Response: Thanks! The information has been fixed!
- The placement of all the title figures is wrong, it should be under the figure. and provide details of each image obtained from where? what software do you use? what about permissions and licenses to use it?
Response: Thanks! The information has been fixed!
- Figures 4, 6, 7, 8 are not clear, what compounds are they? why is there no indication of the names of each molecule? and where did this image come from? what software do you use? what about permissions and licenses to use it?
Response: Thanks! The information has been fixed! Captions were prepared with the names of the compounds in the figures. All the compound structures were manually drawn using the ChemDraw Professional 16.0 software. The figures were created by the authors, with no need for permissions and licenses to use them.
- Almost everything discussed is seaweeds or algae, but in the title there is "corals" and a few appear and add "corals" comprehensively.
Response: Thanks for the comment! As reported in this article, there are few works on “stony” corals on the Brazilian coast (in the last 5 years), to the detriment of “octocorals”. Such hypotheses may be due to the unsustainable exploration of these marine resources, as well as the reduced production during the COVID-19 pandemic. More information has been inserted in the text!
7. Those are all major and basic problems, if they are not corrected properly, then they will not meet the scientifically soundness criteria!
Response: Thanks for the comment! We would like to thank you for your kindly to let us revise our previous version. We hope that the manuscript is now suitable for publication.

Reviewer 2 Report
Overall this is a review paper discuss and summarise the study of seaweed and coral in Brazilian coast. My comment as follow:
Title: accepted
Introduction:
Any references for the fact of seaweed market value and production? Is it an updated information?
Methodology: accepted. if any possible provide few references
Separate results and discussion
Authors are mainly focus on seaweeds. How about coral?
Table 1 and 2 can be enhanced by provided the application of the mentioned seaweed
Need to proofread again
Author Response
Dear Reviewer #2:
Overall this is a review paper discuss and summarise the study of seaweed and coral in Brazilian coast. My comment as follow:
- Title: accepted.
Response: The title was changed as suggested by reviewer 1: "Seaweeds and corals from the Brazilian coast: Review on biotechnological potential and environmental aspects”!
- Introduction: Any references for the fact of seaweed market value and production? Is it an updated information?
Response: Thanks for observation! Reference 5 has been inserted to strengthen the information in the text!
- Methodology: accepted. if any possible provide few references.
Response: Thanks! The electronic sites of the databases have been inserted!
- Separate results and discussion
Response: Thanks for the sugestion! However, as this is a review, we chose to keep the sections together!
- Authors are mainly focus on seaweeds. How about coral?
Response: Thanks for the comment! As reported in this article, there are few works on “stony” corals on the Brazilian coast (in the last 5 years), to the detriment of “octocorals”. Such hypotheses may be due to the unsustainable exploitation of these marine resources, as well as the reduced production during the COVID-19 pandemic. More information has been inserted in the text!
- Table 1 and 2 can be enhanced by provided the application of the mentioned seaweed
Response: Thanks! Applications of seaweed are reported in tables 1 and 2!

Reviewer 3 Report
In this review the authors described the biotechnological potential and environmental aspects of seaweeds and corals from the Brazilian coast. Using databases analyses they reported the secondary metabolites and biological activities of seventy[1]one seaweeds species from the Chlorophyta, Rhodophyta, and Ochrophyta phyla and fifteen corals. Also they described the environmental dangers to humans and environment related to these species and the gaps in the research that are missing.
This topic is relevant to the field because it gives a wide and comprehensive overview of the biotechnological potential of seaweeds and corals in Brazil together with potential negative environmental impact. Those are rather low number of analysis of bioactive molecules and from the environmental point analysis of presence of microplastics in particular.
Compared with other similar published material this study adds a comprehensive view and analysis of impact on environment related to the seaweeds and corals that is often lacking.
The authors used adequate methods (data bases) in this study to analyze the subject of the study. The methods are also sufficiently described.
The discussion section is well organized and detailed and formed conclusions are appropriate and based on the data obtained in the study and answer the questions posed by the study.
References used by authors in the review are relevant and sufficient.
Also, tables and figures are well presented and descriptive fitting the body and narrative of the text.
Author Response
Dear Reviewer #3:
- In this review the authors described the biotechnological potential and environmental aspects of seaweeds and corals from the Brazilian coast. Using databases analyses they reported the secondary metabolites and biological activities of seventy[1]one seaweeds species from the Chlorophyta, Rhodophyta, and Ochrophyta phyla and fifteen corals. Also they described the environmental dangers to humans and environment related to these species and the gaps in the research that are missing.
Authors' comments: Thanks for the comments and compliments on our article!
- This topic is relevant to the field because it gives a wide and comprehensive overview of the biotechnological potential of seaweeds and corals in Brazil together with potential negative environmental impact. Those are rather low number of analysis of bioactive molecules and from the environmental point analysis of presence of microplastics in particular.
Authors' comments: Thanks for the comments and compliments on our article!
- Compared with other similar published material this study adds a comprehensive view and analysis of impact on environment related to the seaweeds and corals that is often lacking.
Authors' comments: Thanks for the comments and compliments on our article!
- The authors used adequate methods (data bases) in this study to analyze the subject of the study. The methods are also sufficiently described.
Authors' comments: Thanks for the comments and compliments on our article!
- The discussion section is well organized and detailed and formed conclusions are appropriate and based on the data obtained in the study and answer the questions posed by the study.
Authors' comments: Thanks for the comments and compliments on our article!
- References used by authors in the review are relevant and sufficient.
Authors' comments: Thanks for the comments and compliments on our article!
- Also, tables and figures are well presented and descriptive fitting the body and narrative of the text.
Authors' comments: Thanks for the comments and compliments on our article!
